# IDSpace: A Model-Guided Synthetic Identity Document Generation Framework and Dataset

## Abstract

To address the challenges in the lack of data for evaluating identity document fraud detection models provided by vendors or merchants, we propose IDSpace, a cost-effective framework for generating high-quality synthetic identity documents. Our IDSpace framework can generate a large number of identity documents using only a small number of documents from the target domain, while overcoming limitations imposed by privacy constraints and ensuring that the evaluation results using our synthetic images are consistent with images from the target domain. Our framework also allows advanced users to flexibly specify the metadata regarding the entities, capturing devices, and backgrounds of the documents to be generated. To achieve these benefits, IDSpace has introduced two key innovations: (1) abstracting the synthetic data generation process as a function of control parameters and metadata, and thus decoupling the user-centric metadata customization process and the automatic parameter tuning process; and (2) a model-guided few-shot document generation methodology that employs Bayesian optimization (BO) to align generated documents with the target domain, ensuring fidelity and utility for model evaluation using minimal samples from the target domain.

## 1 Introduction

The surge in digital platforms offering remote identity proofing has heightened concerns about the forgery of identity documents, such as passports, driver's licenses, and identity cards. In fiscal year 2023, the Financial Crimes Enforcement Network received approximately 4.6 million Suspicious Activity Reports, with around 1.75 million of those reports related to identity fraud aba.com (2024). Accurate detection of fraudulent identity documents is crucial for reducing authentication risks across various sectors, including finance, healthcare, travel, retail, government, and gambling Onfido (2023). However, due to the sensitivity of personal information in these documents, assembling a comprehensive real-world dataset to evaluate existing fraud detection models is highly challenging.

**A motivating example.** Between August 2023 and April 2024, the US General Services Administration (GSA) conducted a large-scale study, recruiting 3,991 participants from five racial and ethnic demographic groups Fatima et al. (2024). Participants were instructed to capture and upload images of their identity documents and the corresponding selfies using mobile devices, allowing the evaluation of five commercial remote identity verification services. Obviously, this methodology is time-consuming, costly, and offers limited control over the quality and distribution of the collected images. Furthermore, the involvement of human subjects restricts its generalizability.

The recent advances in AI/ML have spurred interest in generating synthetic identity documents to augment training data for fraud detection models Arlazarov et al. (2019); Bulatovich et al. (2022); Guan et al. (2024). However, existing approaches exhibit the following limitations that hinder their applicability to our target scenario–evaluating the performance of fraud detection models:

• **The lack of balance between cost/feasibility and quality.** (1) Manual generation of identity document datasets—such as MIDV-500 Arlazarov et al. (2019) and MIDV-2020 Bulatovich et al. (2022), is expensive and labor-intensive, leading to limited scale. For example, MIDV-2020 Bulatovich et al. (2022) contains only 1,000 distinct template images across ten document categories. (2) Training-based approaches Benalcazar et al. (2023), including those based on Generative Adversarial Networks (GANs) Goodfellow et al. (2020) and differential privacy Dwork (2006), require large volumes of real identity documents from the target domain, which are difficult to obtain due to

regulatory constraints. (3) Few-shot approaches Bothra et al. (2023); Lerouge et al. (2024); Xie et al. (2024) aim to minimize data requirements by using inpainting techniques to fill in synthetic personal data and face images into a template (illustrated as Template (T) in Fig. 1). While these methods are cost-effective and improve training data diversity, they often produce content that diverges from the target domain distribution. This domain shift can mislead fraud detection models and result in inconsistent or unreliable evaluation outcomes.

• **The lack of flexible user-level control.** Comprehensive benchmarking and evaluation of fraud detection models require testing under a wide range of conditions and edge cases, such as documents from a minority demographic group. To support such nuanced evaluations, it is essential to provide a flexible, user-centric interface that allows end-users to specify the characteristics of the document they wish to generate. Such control enables targeted synthesis of evaluation data tailored to specific benchmarking scenarios.

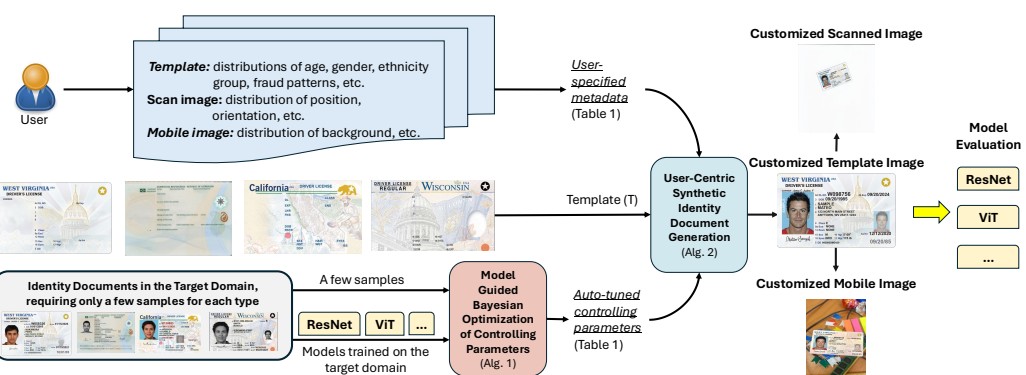

Figure 1: Overview of Our Synthetic Template Image Generation.

To close the gaps, we propose a cost-effective framework based on the few-shot approach, called IDSPACE, for generating identity documents, including customized template documents, scanned documents, and documents captured by mobile phones, to accurately and flexibly evaluate targeting fraud detection models, as illustrated in Fig. 1. Our key contributions include:

• We proposed a novel problem abstraction that decouples the metadata that users want to specify (e.g., age, gender, and ethnicity group of the entity), and auto-tuned control parameters that determine how the user-specified metadata are integrated with the template (T) to compose the final output documents. Usually, those control parameters (e.g., font style and font size for each text field, and noise, brightness, contrast, and sharpness, which varies for different types of scanner or mobile phone device configuration, are hard to customize for ordinary users. Different applications require distinguishing between metadata and control parameters in different ways. For example, when the application needs to evaluate fraud detection tasks on documents captured with low image quality, it considers image quality parameters (e.g., the image quality level of the JPEG file, subsampling, and q-tables Rabbani & Joshi (2002)) as user-customizable metadata. However, if all documents in the target domain are expected to be captured using similar configurations, auto-tuning these parameters makes more sense.

• We also demonstrate that the above flexible abstraction leads to a novel cost-effective few-shot approach: We only need a small number (e.g., around or less than 40) of documents from the target domain to automatically tune the control parameters to obtain high-quality generated documents using a novel model-guided Bayesian optimization (BO) methodology. By leveraging a small set of samples in the target domain and the target model(s) for measuring the model evaluation consistency, our method generates synthetic evaluation data that aligns closely with the performance characteristics observed on the target domain. The cost-effectiveness is critical to our targeting scenarios where privacy or legal constraints limit access to real documents in the target domain.

• We provide a large-scale identity document dataset with examples illustrated in Fig. 2, publicly released on `https://huggingface.co/datasets/Anonymous-111/IDSPACE` (under CC-BY 4.0). Our evaluation further showed that IDSPACE achieved prediction results that are consistent with corresponding documents from the target domain for different fraud detection models, making it effective for benchmarking fraud detection tasks, which may extend to other tasks. (The source code of our IDSPACE

framework is submitted as part of the supplementary material.We will open-source the code base under the Apache 2.0 license once the paper is accepted.) We further demonstrate that the dataset generated by IDSPACE also benefits model training. The scalability and flexibility of our generation pipeline make it suitable for training and evaluating modern deep-learning architectures, especially in low-resource or privacy-sensitive settings.

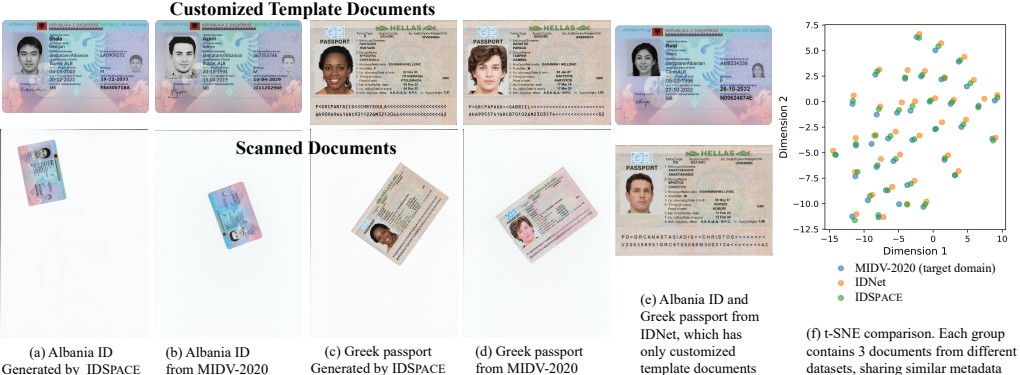

Figure 2: Examples and t-SNE comparison for non-fraud documents generated by our IDSPACE, IDNet Xie et al. (2024), and MIDV-2020 (the target domain). (More examples are in the Appendix.)

## 2 RELATED WORK

**Public Synthetic Datasets and Synthetic Data Generation Methods for Identity Documents.** As mentioned, manually crafted datasets such as **MIDV-500** Arlazarov et al. (2019), **MIDV-2020** Bulatovich et al. (2022), and **KID34K** Park et al. (2023) suffer from the document generation cost, limited to around 100 templated documents for one country. **SIDTD** Boned et al. (2024) and **FMIDV** Al-Ghadi et al. (2023) rely on MIDV-2020 to provide fraud patterns using inpainting or crop-and-move techniques. Overall, these datasets lack diversity and flexibility in conducting comprehensive and customizable model evaluation tasks. Other large-scale datasets suffer from different quality issues, which fall into the following categories. **(1) Redaction-based document generation.** For example, **BID** de Sá Soares et al. (2020) redacts sensitive information such as portrait photos from real-world documents, which reduces their utility for many portrait-based fraud detection applications, e.g., face morphing detection. **(2) Training-based generation Benalcazar et al. (2023)** Generative-Adversarial Network (GAN) is widely used for synthetic document generation. For example, **StyleGAN2** Karras et al. (2020) was used to generate ID images due to its strong capability in synthesizing highly realistic human faces. However, it struggled with alphanumeric characters, and it also needed thousands of real IDs for model training, which is impractical in our target scenarios where only a small number of documents from the target domain of models to be evaluated is available. **(3) Domain adaptation techniques** Sugiyama et al. (2007); Tzeng et al. (2017). For example, **CycleGAN** Xie et al. (2020) can be used to adapt low-fidelity identity documents to the target domain. However, this approach usually requires a large number of labeled samples of the target domain, which is not practical in our target scenarios. Ben-David et al. David et al. (2010) proposed impossibility theorems, indicating that even with small distributional divergence or a universally good classifier, domain adaptation can fail without labeled target data. **(4) Few-shot Approaches. DocXPand-25k** Lerouge et al. (2024) applies inpainting to fill metadata into a self-designed document template, and it can hardly generalize to real-world document types. **IDNet** Guan et al. (2024); Xie et al. (2024) used Bayesian optimization to tune only a couple of parameters with an objective to maximize the similarity between the generated documents and target domain documents. While it is instrumental in training robust fraud detection models and has been downloaded for more than 8,600 times on Zenodo, we found the target models tend to predict inconsistent results for the documents from the target domain and the IDNet-generated documents. That is due to a discrepancy between the generated dataset and the target domain, as illustrated in Fig. 2(f). In this work, we addressed such limitation by introducing a novel model guiding data generation technique coupled with a user-centric interface that flexibly separate metadata from control parameters.

**Synthetic Data Generation for Model Evaluation.** The use of synthetic data in machine learning has been explored across various domains. Nikolenko et al. (2021) discusses adaptation and refinement

Table 1: Overview of Examplar Synthetic ID Generation Parameters

| | control parameters (AutoTuned) | User-Customizable Metadata |
|---|---|---|
| Customized Template Documents | **Font:** Font style and size of each text field. We support Google's font library, containing 1816 font families. **Text color:** Color of texts that overlap with background. **Positions:** Positions of each field in the template, and character spacing within each field. **Quality:** JPEG image quality settings (e.g., subsampling=4:2:0, qtables=standard, quality factor in [50–95]). | **ID Template:** File path of the template image used for ID generation. **Fraud Pattern:** Whether to simulate fraud and which type—e.g., *crop-and-move*, *inpaint-and-rewrite*, following SIDTD Boned et al. (2024). **Entity Information:** First name, last name, DOB, ID number, portrait photo, eye color, height, weight, issue date, expiration date, etc. |
| Scanned/Mobile Documents | **Noise Level:** Amount of Gaussian noise added—e.g., standard deviation in range [0, 25]. **Subtle Blurring:** Gaussian blur with sigma in range [0.5, 2.0] to simulate out-of-focus scans. **Brightness:** Adjustment factor for image brightness—e.g., randomly sampled from [0.8, 1.2]. **Contrast:** Adjustment factor for image contrast—e.g., randomly sampled from [0.8, 1.2]. **Sharpness:** Strength of sharpening filter—e.g., factor in [0.5, 2.0]. | **ID Template Image:** File path of the ID image (a customized template document) to be scanned. **Resolution:** DPI value, e.g., 200, 300, or 400. **Color Mode:** One of {"color", "grayscale", "black and white"}. **Position/Orientation (for scanned images):** Placement of the ID—e.g., rotated by ±5° or shifted ±10px. **Wear and Tear:** Simulated aging artifacts—e.g., *fold marks*, *scratches*, *stains*, or *faded text*. **Background image (for mobile images):** An image to serve as the background of the ID placement. |

of the synthetic to the real domain using GANs. van Breugel et al. (2023) propose to use a deep generative model on the test dataset to create synthetic data for evaluating model performance on underrepresented subgroups and under distributional shifts. Their work shows that synthetic data generated in this way outperforms real test data in estimating performance for minority subgroups and under shifts, while also providing uncertainty estimates. However, their approach relies on the availability of test data, which is often scarce in practice in our targeting scenarios.

## 3   A NOVEL PROBLEM ABSTRACTION

In this section, we present a few-shot synthetic data generation methodology designed to balance cost and quality in evaluating fraud detection models. Our approach is to combine the decoupling of the user-specified metadata and the control parameters finetuned by a model-guiding framework.

Table 1 illustrates how we distinguish metadata that should be explicitly specified by users from parameters that should be automatically adjusted in this work. Parameters that can be reliably inferred using external tools are excluded. For instance, the background color of a portrait photo can often be extracted using standard color analysis tools and does not require automated tuning. In contrast, detecting text color when it overlaps with complex background images is significantly more difficult, and identifying the original font styles used in official identity documents is often infeasible due to their proprietary nature. In such cases, we rely on auto-tuning. Importantly, advanced users can redefine the boundary between user-specified metadata and automatically controlled parameters, depending on their available tools, expertise, and resources. Based on this flexible and modular design, we are the first to formalize the synthetic identity document generation problem as follows.

**Problem Definition.** Given a template of a type of ID document (e.g., a template of the West Virginia driver's license), denoted as $T$, generating the $i$-th ID image has two steps: (1) obtaining or generating the metadata information $x_{meta}^i$ following user-specification, which consists of fraud patterns, capturing device (e.g., scanner), capturing environments (e.g., rotation and position of the document, color mode, and resolution), and various personal information (e.g., first name, last name, date of birth, ID card number, portrait photo, eye color, height, weight, card issue date, expiration date), listed as user-customizable metadata in Tab. 1. (2) filling in the metadata information into the template to generate the final image $x_i$, denoted as $x_i = G_\theta(x_{meta}^i, T)$. Here, $\theta$ represents the parameters that control filling the metadata into the template, such as those control parameters listed in Tab. 1. While $G_\theta(\cdot)$ represents the process of transforming the metadata $x_{meta}^i$ and the given template $T$ into a synthetic ID image $x_i$ using the parameters $\theta$.

Given an existing machine learning model $f$ trained for fraud detection for a target domain consisting of ID documents sharing the same template $T$, denoted as $\mathcal{D}_{real} = \{x_i\}$, with $f$'s training dataset $\mathcal{D}_{training} \subset \mathcal{D}_{real}$. Assuming each sample $x_i \in \mathcal{D}_{real}$ having metadata $meta(x^i)$, given a small number of samples from $\mathcal{D}_{real}$, we would like to learn $\theta$ so that $\forall x_i \in \mathcal{D}_{real}$, we have $x_i = G_\theta(meta(x_i), T)$, and thus $f(x_i) = f(G_\theta(meta(x_i), T))$.

Examples of templates (T) Xie et al. (2024); Rombach et al. (2022) are illustrated in Fig. 1.

---

Algorithm 1: Model-Guided Bayesian Optimization of Control Parameters (For simplicity, it only used one guiding model $f$, and the extension to multiple guiding models as illustrated in Eq. 1 is trivial)

---

1: **Input:** A small dataset from $\mathcal{D}_{\text{real}}$: $\mathcal{D}_{\text{sample}} = \{x_1, x_2, ..., x_m\}$, a template $T$
2: synthetic data generator $G$, the guiding model $f$, parameter space $\Theta$, weight $\lambda_0, \lambda_1$
3: **Output:** Optimized parameter set $\theta^*$
4: Initialize Bayesian Optimization over $\Theta$          ▷ Setup BO framework
5: **for** iteration $t = 1$ to $T$ **do**
6:     $\theta_t \leftarrow$ BO acquisition function          ▷ Sample candidate parameters
7:     $\mathcal{D}_{\text{syn}} \leftarrow \{G_{\theta_t}(meta(x_i), T)|x_i \in \mathcal{D}_{sample}\}$          ▷ Generate synthetic data
8:     $\mu_{\text{similarity}}^t \leftarrow \frac{1}{m} \sum_{i=1}^m similarity(x_i, D_{\text{syn}}^{(i)})$          ▷ Compute similarity, e.g. SSIM (Eq. 2)
9:     $\mathbf{p}_{\text{real}} \leftarrow f(\mathcal{D}_{\text{sample}})$          ▷ Get model prediction results
10:    $\mathbf{p}_{\text{syn}} \leftarrow f(\mathcal{D}_{\text{synth}})$
11:    $\mu_{\text{consistency}}^t \leftarrow \frac{1}{m} \sum_{i=1}^m \mathbb{1}(p_{\text{real}}^{(i)} = p_{\text{syn}}^{(i)})$          ▷ Compute consistency (Eq. 3)
12:    $J(\theta_t) \leftarrow \lambda_0 \mu_{\text{similarity}}^t + \lambda_1 \mu_{\text{consistency}}^t$          ▷ Compute Objective score (Eq. 1)
13:    Update BO model with $(\theta_t, J(\theta_t))$
14: **end for**
15: **return** $\theta^* = \arg\max_{\theta \in \{\theta_1, ..., \theta_T\}} J(\theta)$          ▷ Select best parameters

---

## 4   MODEL GUIDED AUTOMATIC TUNING OF CONTROL PARAMETERS

To address the problem formalized in Sec. 3, we propose combining Bayesian optimization Frazier (2018) with our custom optimization objective that introduces $l$ guiding models $f_1, \ldots, f_l$, which are trained in the target domain. These $k$ models are not necessarily the target models to be evaluated. The objective is not only to maximize the overall similarity (e.g., measured using structural similarity index measure (SSIM)) between each input document $x_i$ and the corresponding generated document $G_\theta(meta(x_i), T)$ for $i = 1, ..., m$ (See Eq. 2), but also to improve evaluation consistency between $f_k(x_i)$ and $f_k(G_\theta(meta(x_i), T))$ for $i = 1, ..., m$ and $k = 1, ..., l$ (See Eq. 3). Here, $m$ denotes the number of samples from the target domain, which is assumed to be small, given the sensitive nature of ID data. Our custom optimization objective is formalized in Eq. 1, which is a weighted sum of the similarity metric (Eq. 2) and the evaluation consistency metric (Eq. 3).

$$\theta^* = \arg\max_{\theta \in \Theta}(\lambda_0 \cdot \mu_{\text{similarity}}(\theta) + \lambda_1 \cdot \mu_{\text{consistency}}^1(\theta) + \cdots + \lambda_l \cdot \mu_{\text{consistency}}^l(\theta)) \tag{1}$$

$$\mu_{\text{similarity}}(\theta) = \frac{1}{m} \sum_{xi=x_1,...,x_m \in D_{real}} similarity(x_i, G_\theta(meta(x_i), T)). \tag{2}$$

$$\mu_{\text{consistency}}^k(\theta) = \frac{1}{m} \sum_{xi=x_1,...,x_m \in D_{real}} \mathbb{1}(f_k(x_i) = f(G_\theta(meta(x_i), T))), k = 1, \ldots, l \tag{3}$$

Our proposed Bayesian optimization algorithm formalized in Alg. 1 (1) trains a surrogate model to learn the relationship between $\theta$ (the control parameters of the data generation process) and an objective metric as formalized in Eq. 1, (2) iteratively selects $\theta$ guided by the surrogate model to optimize the objective, and (3) in each iteration, evaluates the effectiveness of the selected $\theta$ using the objective function, and use the measured results to update the surrogate model. The Bayesian optimization strategy can be replaced with other search methods, such as Hyperband Li et al. (2018), which are evaluated and compared to our approach in Sec. 6.3.

## 5   USER-CENTRIC SYNTHETIC IDENTITY DOCUMENT GENERATION PROCESS

After optimizing the control parameters using Alg. 1, we apply a synthetic data generation process formalized in Alg. 2. It consists of two steps:

**Step 1. Generating metadata following user specification** (line 3 in Alg. 2). By customizing the metadata, users can control which aspects of the model performance to focus on during the evaluation process. For example, if a user needs to evaluate the model's fraud detection fairness on different gender, age, or ethnicity groups, s/he may request to generate documents with uniform distributions across all groups, including minority groups. Users can specify metadata by either uploading a CSV file that lists metadata for each document or a JSON file that defines the probability distribution of each type of metadata, used as $\theta_{user}$ in Alg. 2. Portrait Photos are selected based on metadata.

---

**Algorithm 2: User-Centric Synthetic Data Generation**

---

1: **Input:** User-specified metadata: $\theta_{user}$; control parameters tuned using Alg. 1: $\theta^*$; ID template: $T$; Synthetic data generator: $G$; Metadata generator: $F$
2: **Output:** Generated document $x_{syn}$
3: $meta \leftarrow F_{\theta_{user}}()$            ▷ Generate metadata based on user specification
4: **return** $x_{syn} = G_{\theta^*}(meta, T)$          ▷ Generate data using tuned control parameters

---

**Step 2. Generating the documents** (line 4 in Alg. 2). Customized template documents, as shown in Fig. 1 and Fig. 2, emphasize geometric precision in text alignment and compliance with governmental design specifications. Such quality requirements are satisfied via the control parameters ($\theta^*$) tuned by our model-guided Bayesian optimization algorithm and the function ($G_{\theta^*}(\cdot)$) that applies those parameters to drive varying-scale synthesis by filling in the template with user-specified personal information and fraud patterns. (Details about the fraud patterns supported in this work can be found in Appendix Sec. A.10.1.) The function ($G_{\theta^*}(\cdot)$) needs to be engineered for each type of document, involving recognizing all fields to be customized, and mapping each field to the corresponding user-provided metadata (i.e., information to be filled) and control parameters (i.e., controlling how to fill the information into the fields). However, this is a one-time cost, which is amortized to the cost-effective generation of unlimited high-quality documents (see Appendix Sec. A.6). In addition to the generated document, each output includes annotations such as typographic specifications, positional coordinates, semantic segment masks, and bounding boxes in JSON format, facilitating downstream tasks, e.g., constructing scanned or mobile documents, as shown in Fig. 1.

Taking scanned documents as an example, the pipeline first loads the user-specified customized template document. Then, it invokes a function to apply both of the tuned control parameters and user-specified metadata to transform the template image. For example, if the customized template document contains Machine Readable Zone (MRZ), our pipeline will apply slight sharpening or contrast enhancement to that specific region. It also masks areas containing security features, such as holograms, to avoid excessive blurring or color alteration. In addition, following the user specification on wear and tear, it will introduce subtle imperfections such as scratches, fading, or corner rounding. Then, the function generates the scanner background image and combines the transformed customized template document and the background following the user-specified position and rotation parameters. The function also adds subtle shadows to create a more realistic appearance. The generation of mobile documents is similar, which we discuss in Appendix Sec. A.13.

## 6 EVALUATION

We conducted a comprehensive empirical study to investigate the following research questions. R1. Will models trained on the target domain achieve consistent evaluation results on our generated document and the corresponding document from the target domain with the same metadata? Will the evaluation consistency of our generated documents outperform alternatives? R2. Does the dataset generated by IDSPACE also benefit the learning process? R3. How is the Bayesian optimization-based parameter tuning approach compared to alternatives?

**System Environment.** All experiments were conducted on an Ubuntu Linux server equipped with 48 CPU cores (Intel Xeon Silver 4310, 2.10 GHz), 125 GB of RAM, and two NVIDIA A10 GPUs (24GB VRAM each). The server is installed with 256GB NVMe SSD and 1TB HDD drive.

**Experiment Setup.** We used the MIDV-2020 and SIDTD datasets, which are under the CC BY-SA 2.5 and CC-BY-4.0 licenses, as the target domain. They include ten types of European country identity documents, with 100 templated documents and around 100 fraud templated documents in each type. We sampled $50\%$ of documents to form a *training set* for learning the fraud detection models, which are used as the guiding models for parameter tuning and the target models to be evaluated. $30\%$ of samples form the *testing set* to evaluate the consistency of the prediction of the model of all the baseline data generation methods. The additional $20\%$ of the documents compose a *tuning set* to adjust the control parameters following Alg. 1. Due to space limitations, our evaluation focuses on fraud detection using customized template images, which is fundamental to identity fraud detection Boned et al. (2024); MIDV-2020; Guan et al. (2024). We put the corresponding evaluation results for scanned documents in the Appendix. In our implementation of Alg. 1, we used SSIM as our similarity metric. We set $\lambda_0$ to 1 and $\lambda_i$ to $1/l$ ($l$ is the number of guiding models, $i = 1, \ldots, l$)

to balance similarity and consistency. The tuning of $\lambda_i$ is discussed in the Appendix. Due to space limitations, we mainly showed the results on the Albania ID documents. The results on other types of documents are similar, also detailed in the Appendix.

## 6.1 R1. MODEL PREDICTION CONSISTENCY COMPARISON

To answer R1, following existing works on fraud detection for identity documents Boned et al. (2024); Park et al. (2023); Onfido (2023); Bayer et al. (2025); Khare et al. (2024); Mahadevan et al. (2023); Bruveris et al. (2020); Gietema et al. (2024), we used five fraud detection models with varying model architectures, including ViT-large, ResNet50, Inception-v3, VGG16, and DenseNet trained from the target domain, achieving accuracies of 1.0000, 1.0000, 1.0000, 0.9873, and 1.0000, respectively (see Appendix Sec. A.12 for more details about those models' usage on identity document fraud detection).

We then use each of the following baselines with the *tuning set* to generate a dataset, using the same metadata as the samples from the *testing set* from the target domain. We further compare the model prediction consistency, defined in Eq. 3, between the dataset generated by each baseline and the *testing set* for each of the five models.

• **BO w/ SSIM-only objective**: This is a variation of IDSPACE without any model guidance, i.e., using Eq. 2 as the objective function of the Bayesian Optimization (BO) search.

• **CycleGAN**: This is a widely used domain adaptation approach with pre-trained models publicly available Zhu et al. (2017); CycleGAN under BSD license. We chose the CycleGAN model pretrained on the Flickr dataset Flickr, and finetuned it using the *tuning set* to adapt an ID dataset generated by BO w/ SSIM-only objective to the target domain.

• **IDSPACE**: This is our approach using different combinations of guiding models for BO search.

The comparison results are presented in Tab. 2. We observed significant improvement of consistency scores using our proposed model-guided Bayesian optimization methodology, ranging from 15.62% to 30.55% compared to BO w/ SSIM-only objective, and 27.40% to 42.33% compared to CycleGAN. We also found that using our proposed approach, the consistency is not only improved for the guiding models used in the BO objective function, but also improved for other models trained on the target domain. In addition, incorporating more guiding models further improved consistency in the majority of the cases. The algorithm demonstrated robustness across different architectures (mean consistency = 0.9354±0.003%). Using our approach, small models such as Inception-v3 and DenseNet are easier to achieve better consistency than other larger models.

To investigate how the model prediction consistency changes with the number of samples available from the target domain for different baselines, we applied 2, 20, and 40 samples (with balanced fraud and non-fraud labels) from the *tuning set* to finetune the CycleGAN model, and to tune the control parameters for the baseline using BO w/ SSIM-only objective and our model-guided IDSPACE approach. We used the *testing set* that is disjoint with the *tuning set* to measure the prediction consistency of the ResNet50 model used in Tab. 2 between the documents from the *testing set* and the generated documents, both of which share the same metadata (i.e., field values and photos). Our IDSPACE approach used the target ResNet50 model as the guiding model.

The results in Tab. 3 showed that our approach consistently and significantly improved consistency even when fewer samples are available from the target domain. This result demonstrated the benefits of incorporating guiding models into the search objective for a few-shot approach. Furthermore, the overall objective function improved as the number of samples from the target domain increased. Notably, even with just two samples, our algorithm demonstrated substantial improvement compared to baselines, proving the effectiveness of our proposed model-guided framework.

**Discussions on Other Baselines.** We trained StyleGAN Karras et al. (2020) using different amounts of documents from MIDV and SIDTD, and found that the available documents in these datasets are insufficient to obtain reasonable outputs. We also investigated the zero-shot and few-shot capabilities of existing generative AI models, such as gpt-image-1 from OpenAI Chatgpt, and identified that the documents generated by those tools have a poor visual fidelity. In addition, fine-tuning or adapting these models for synthetic document generation requires many documents from the target domain,

Table 2: Model prediction consistency evaluation on different models with best results highlighted in blue

| | Consistency with Target Test Data (Mean ± Std) | | | | | Average(row) |
|---|---|---|---|---|---|---|
| | ViT-Large | ResNet50 | Inception-v3 | VGG16 | DenseNet | |
| CycleGAN | 0.5648 ± 0.004 | 0.5324 ± 0.000 | 0.5093 ± 0.000 | 0.5648 ± 0.000 | 0.5000 ± 0.000 | 0.5343 ± 0.001 |
| BO w/ SSIM-only objective | 0.7257 ± 0.018 | 0.5382 ± 0.006 | 0.8646 ± 0.012 | 0.6146 ± 0.006 | 0.5174 ± 0.012 | 0.6521 ± 0.017 |
| Guiding models | IDSPACE | | | | | |
| DenseNet | 0.8819 ± 0.093 | 0.8646 ± 0.018 | 0.9965 ± 0.006 | 0.8715 ± 0.040 | 1.0000 ± 0.000 | 0.9229 ± 0.005 |
| DenseNet + Inception-v3 | 0.8021 ± 0.136 | 0.8750 ± 0.017 | 0.9931 ± 0.007 | 0.8715 ± 0.060 | 1.0000 ± 0.000 | 0.9083 ± 0.007 |
| DenseNet + Inception-v3 + ResNet50 | 0.8194 ± 0.091 | 0.9306 ± 0.017 | 1.0000 ± 0.000 | 0.8889 ± 0.028 | 1.0000 ± 0.000 | 0.9278 ± 0.006 |
| DenseNet + Inception-v3 + VGG16 | 0.8924 ± 0.015 | 0.8646 ± 0.036 | 1.0000 ± 0.000 | 0.9410 ± 0.025 | 1.0000 ± 0.000 | 0.9396 ± 0.004 |
| DenseNet + Inception-v3 + ViT-Large | 0.9306 ± 0.020 | 0.8889 ± 0.045 | 0.9965 ± 0.006 | 0.9097 ± 0.036 | 1.0000 ± 0.000 | 0.9451 ± 0.003 |
| DenseNet + ResNet50 | 0.9340 ± 0.015 | 0.9236 ± 0.016 | 1.0000 ± 0.000 | 0.8819 ± 0.035 | 1.0000 ± 0.000 | 0.9479 ± 0.003 |
| DenseNet + ResNet50 + VGG16 | 0.9236 ± 0.029 | 0.8924 ± 0.006 | 1.0000 ± 0.000 | 0.9375 ± 0.016 | 1.0000 ± 0.000 | 0.9507 ± 0.002 |
| DenseNet + Resnet50 + ViT-Large | 0.9201 ± 0.033 | 0.9062 ± 0.041 | 0.9792 ± 0.036 | 0.8403 ± 0.094 | 0.9965 ± 0.006 | 0.9285 ± 0.004 |
| DenseNet + VGG16 | 0.9132 ± 0.012 | 0.8819 ± 0.029 | 1.0000 ± 0.000 | 0.9306 ± 0.034 | 1.0000 ± 0.000 | 0.9451 ± 0.003 |
| DenseNet + VGG16 + ViT-Large | 0.9306 ± 0.017 | 0.8542 ± 0.030 | 1.0000 ± 0.000 | 0.9479 ± 0.032 | 1.0000 ± 0.000 | 0.9465 ± 0.004 |
| DenseNet + ViT-Large | 0.9306 ± 0.039 | 0.8646 ± 0.060 | 1.0000 ± 0.000 | 0.8958 ± 0.029 | 1.0000 ± 0.000 | 0.9382 ± 0.004 |
| Inception-v3 | 0.6146 ± 0.108 | 0.7778 ± 0.057 | 0.9931 ± 0.007 | 0.7118 ± 0.129 | 0.9444 ± 0.056 | 0.8083 ± 0.025 |
| Inception-v3 + ResNet50 | 0.7708 ± 0.068 | 0.9340 ± 0.030 | 0.9965 ± 0.006 | 0.8681 ± 0.012 | 0.9965 ± 0.006 | 0.9132 ± 0.009 |
| Inception-v3 + ResNet50 + VGG16 | 0.9375 ± 0.023 | 0.9097 ± 0.025 | 0.9965 ± 0.006 | 0.9479 ± 0.027 | 0.9965 ± 0.006 | 0.9576 ± 0.001 |
| Inception-v3 + ResNet50 + ViT-Large | 0.9375 ± 0.007 | 0.9271 ± 0.027 | 0.9965 ± 0.006 | 0.8785 ± 0.027 | 1.0000 ± 0.000 | 0.9479 ± 0.003 |
| Inception-v3 + VGG16 | 0.9375 ± 0.030 | 0.8750 ± 0.026 | 1.0000 ± 0.000 | 0.9549 ± 0.025 | 1.0000 ± 0.000 | 0.9535 ± 0.003 |
| Inception-v3 + VGG16 + ViT-Large | 0.9410 ± 0.030 | 0.8194 ± 0.033 | 1.0000 ± 0.000 | 0.9410 ± 0.025 | 1.0000 ± 0.000 | 0.9403 ± 0.005 |
| Inception-v3 + ViT-Large | 0.9340 ± 0.023 | 0.8576 ± 0.021 | 0.9965 ± 0.006 | 0.9062 ± 0.032 | 1.0000 ± 0.000 | 0.9389 ± 0.004 |
| ResNet50 | 0.9271 ± 0.015 | 0.9514 ± 0.012 | 1.0000 ± 0.000 | 0.8993 ± 0.023 | 1.0000 ± 0.000 | 0.9556 ± 0.002 |
| ResNet50 + VGG16 | 0.9097 ± 0.016 | 0.9097 ± 0.029 | 1.0000 ± 0.000 | 0.9306 ± 0.014 | 1.0000 ± 0.000 | 0.9500 ± 0.002 |
| ResNet50 + VGG16 + ViT-Large | 0.9375 ± 0.012 | 0.9167 ± 0.026 | 1.0000 ± 0.000 | 0.9132 ± 0.006 | 1.0000 ± 0.000 | 0.9535 ± 0.002 |
| ResNet50 + ViT-Large | 0.9132 ± 0.023 | 0.9340 ± 0.050 | 0.9965 ± 0.006 | 0.8785 ± 0.032 | 1.0000 ± 0.000 | 0.9444 ± 0.003 |
| VGG16 | 0.9271 ± 0.040 | 0.8368 ± 0.054 | 0.9965 ± 0.006 | 0.9306 ± 0.052 | 1.0000 ± 0.000 | 0.9382 ± 0.004 |
| VGG16 + ViT-Large | 0.9514 ± 0.021 | 0.8542 ± 0.025 | 1.0000 ± 0.000 | 0.9444 ± 0.017 | 1.0000 ± 0.000 | 0.9500 ± 0.004 |
| ViT-Large | 0.9375 ± 0.016 | 0.8646 ± 0.065 | 0.9861 ± 0.010 | 0.8750 ± 0.024 | 0.9965 ± 0.006 | 0.9319 ± 0.004 |
| Average(Column) | 0.8982 ± 0.006 | 0.8846 ± 0.002 | 0.9969 ± 0.000 | 0.8999 ± 0.003 | 0.9972 ± 0.000 | |

Table 3: Comparison of consistency with ResNet50 trained in the target domain for different baselines with best results highlighted in blue.

| Samples from Target Domain | CycleGAN | BO w/ SSIM-only objective | IDSPACE |
|---|---|---|---|
| 2 | 0.5463 ± 0.000 | 0.4132 ± 0.000 | 0.8056 ± 0.008 |
| 20 | 0.5417 ± 0.000 | 0.1701 ± 0.000 | 0.9444 ± 0.000 |
| 40 | 0.5324 ± 0.000 | 0.5382 ± 0.006 | 0.9514 ± 0.012 |

which is infeasible in our target use case, given the sensitivity of the identities. More detailed analysis is presented in Appendix Sec. A.7 and Sec. A.9.

## 6.2 R2. FRAUD DETECTION TRAINING ACCURACY ON DIFFERENT SYNTHETIC DATASETS

As demonstrated in Sec. 6.1, our proposed approach significantly improved the evaluation consistency of the generated documents with the *testing set*. Next, we will evaluate whether the fraud detection models trained on our generated dataset could also generalize well to the documents in the target domain. We applied the IDSPACE framework to generate a synthetic ID dataset. We only used ResNet50 used in Sec. 6.1 as the guiding model. We used the *tuning set* with 40 documents with balanced labels from the target domain to tune the control parameters. Leveraging these tuned parameters, we then reuse the metadata distribution of the IDNet dataset, which was released on Zenodo in 2024, to generate a dataset with the same number of non-fraud samples, which is 5,979 samples, for each of ten European country's indentity document types contained in the (non-fraud) MIDV-2020 and the (fraud) SIDTD datasets. In addition, we generate 5,979 samples for each of two fraud patterns (i.e., crop-and-move, inpainting and rewriting) proposed by the SIDTD benchmark, and also included in the IDNet dataset, detailed in Appendix Sec. A.10.1. To compare the utility of the generated dataset to the IDNet dataset, we trained different fraud detection models on each dataset and evaluated the utility of these models using the evaluation dataset from MIDV-2020/SIDTD (i.e., the target domain). The results are shown in Tab. 4, which demonstrates the excellent utility of our IDSPACE framework in learning fraud detection tasks in the target domain.

## 6.3 R3. COMPARISON OF PARAMETER TUNING APPROACHES

In this section, we compare our model-guided Bayesian optimization (BO) method with Hyperband, which accelerates the search for optimal configurations by adaptively allocating resources to promising

|  | Target Models | | | |
|---|---|---|---|---|
|  | ResNet50 | Inception-v3 | DenseNet | EfficientNet |
| IDNet | $0.9156 \pm 0.016$ | $0.9536 \pm 0.006$ | $0.8734 \pm 0.012$ | $0.9873 \pm 0.000$ |
| IDSPACE | $0.9866 \pm 0.000$ | $0.9958 \pm 0.000$ | $0.9536 \pm 0.006$ | $0.9958 \pm 0.000$ |

Table 4: Utility (detection accuracy of the copy-and-move frauds) of the models trained on IDNet and IDSPACE, w/ best values highlighted in blue.

Figure 3: Bayesian search vs. Hyperband search

candidates using early-stopping and successive halving Li et al. (2018). For both approaches, we used $40$ (i.e., $20\%$) samples from the *tuning set*, and ResNet50 served as the guiding and the target model. In BO, hyperparameters *init_point* and *n_iter* control the accuracy vs latency tradeoff. Similarly, in Hyperband search, *max_resources* controls the maximum amount of resources that can be allocated to a single configuration, and $\eta$ controls the proportion of configurations discarded in each round of successive halving. Let us denote each instance of $(init\_point, n\_iter)$ pairs and $(max\_resources, \eta)$ pairs as $b_i$ and $h_i$, respectively. In Fig. 3, the evaluated $\{b_1, b_2, ..., b_6\}$ (green points) for BO are $\{(50, 100), (50, 150), (50, 200), (100, 400), (100, 600), (100, 800)\}$, and $\{h_1, h_2, ..., h_6\}$ (red points) for Hyperband search include $\{(500, 3), (700, 3), (500, 2), (900, 2), (1000, 2), (1050, 2)\}$. As shown in the figure, BO outperforms Hyperband search in terms of both tuning latency and consistency score. We observe that BO achieves the peak consistency score of $0.95$ for $b_3$, which took $48.9$ minutes, while the Hyperband search reached its maximum consistency score of $0.85$ for $h_5$, taking $280.15$ minutes. Therefore, BO outperforms the Hyperband by $5.7\times$ in terms of tuning latency while the peak consistency score achieved by Bayesian optimization is $11.76\%$ better than the best consistency score of Hyperband search.

# 7 OUR IDSPACE DATASET AND CONCLUSIONS

**Our IDSPACE Dataset.** As mentioned in Sec. 6.2, we generated 5,979 non-fraud documents, and 5,979 fraud documents for each fraud pattern, for each of ten European identity document types. Moreover, for each of these documents, we also generated one scanned document using randomly selected positions and rotations, using IDSPACE. In total, our new dataset consists of 179,370 customized template documents, 179,370 scanned documents, and a small set of 500 mobile documents, with details described in Sec. A.13. More examples are shown in Appendix Sec. A.1 and Sec. A.13. We published our dataset on HuggingFace https://huggingface.co/datasets/ Anonymous-111/IDSPACE for public access. We discussed and evaluated the efforts required to extend IDSPACE to a new type of identity document (e.g., West Virginia Drivers' license) in Appendix Sec. A.6, and the generation stealty in Sec. A.10.2.

**Summary.** The work is motivated by real-world requirements for a flexible and parameterized synthetic identity document generation framework from US General Services Administration and US Department of Homeland Security, where privacy regulations cause the lack of data for evaluating vendors' software. Our experiments demonstrate that **IDSPACE** effectively bridges the gap between synthetic and the target domain of identity documents for fraud detection model evaluation and training. By leveraging **model-guided BO-tuned generation**, the synthetic data preserves the model prediction consistency of documents from the target domain, enabling reliable model evaluation. Additionally, the framework offers a **user-centric interface** that allows users to decouple metadata that can be flexibly specified by users from control parameters that needs to be automatically tuned.

**Impact.** IDSPACE is a novel model-guided synthetic data generation approach, designed to address the shortage of accessible real data for trustworthy, reliable, comprehensive, and flexible evaluation of existing fraud detection models. Furthermore, our Bayesian optimization-based strategy ensures that synthetic data can be aligned with the target domain using only a small set of real samples, significantly reducing the costs and dependency on sensitive real documents for both evaluation and training. Empirical evidence shows that both model evaluation and training conducted on datasets generated using our IDSPACE outperform other baselines.

**Ethics.** Synthetic data plays a critical role in reducing privacy risks, yet we recognize the potential for dual-use. Malicious actors might attempt to misuse our framework for producing counterfeit documents. To mitigate this, we deliberately restrict the realism of generated outputs, ensuring that synthetic IDs do not contain functionally valid elements such as scannable barcodes. In addition, **all**

**portrait photos and ID entity information used in this study are** $100\%$ **synthetically generated**. The portrait photos are collected from a public synthetic dataset for academic research Photos.

**Reproducibility Discussion.** In our code base (submitted as part of the supplementary material), we provided detailed documentation and automated Python scripts to ensure reproduction. The data required for reproduction has been uploaded to `https://huggingface.co/datasets/Anonymous-111/IDSPACE`

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

## A    TECHNICAL APPENDICES AND SUPPLEMENTARY MATERIAL

### A.1    MORE DOCUMENT EXAMPLES.

Figures 4 to 13 illustrate sample identity documents from datasets corresponding to 10 countries: Albania (ALB), Azerbaijan (AZE), Spain (ESP), Estonia (EST), Finland (FIN), Greece (GRC), Latvia (LVA), Russia (RUS), Serbia (SRB), and Slovakia (SVK). The subfigures (a) through (j) in each of the three figures depict the following aspects, and the fraudulent regions are highlighted with red bounding boxes in subfigures (b), (c), (e) and (f):

(a)  Non-fraud template image from the MIDV dataset.

(b)  *Inpaint-and-rewrite* fraud sample based on the MIDV image created by SIDTD dataset.

(c)  *Crop-and-replace* fraud sample based on the MIDV image created by SIDTD dataset.

(d)  Sample from our improved dataset(IDSPACE) generated using SSIM combined with ResNet50 model guidance as the BO objective.

(e)  *Inpaint-and-rewrite* fraud sample using a template generated by IDSPACE.

(f)  *Crop-and-replace* fraud sample using a template generated by IDSPACE.

(g)  Scanned image from the MIDV dataset.

(h)  Simulated scanned version of the samples in MIDV dataset, replicating the scanning artifacts of (d).

(i)  Simulated scanned version of the samples in MIDV dataset, replicating the scanning artifacts of (e).

(j)  Simulated scanned version of the samples in MIDV dataset, replicating the scanning artifacts of (f).

These examples demonstrate the diversity and realism of our generated dataset.

### A.2    CONSISTENCY EVALUATION RESULTS FOR MORE TYPES OF CUSTOMIZED TEMPLATE DOCUMENTS.

We applied our Bayesian Optimization (BO) search-based approach using different combinations of guiding models to generate customized template documents of othercountries, such as Finland, with the same experimental settings introduced in Section 6. The comparison results are shown in Table 5. Similar to the results presented in Table 2, we observe a significant improvement in consistency scores when using our proposed model-guided BO methodology, compared to the method that uses SSIM as the sole objective (first row in the table). Specifically, our model-guided BO approach achieves relative improvements in consistency scores ranging from $22.00\%$ to $34.86\%$ over the SSIM-only baseline.

### A.3    CONSISTENCY EVALUATION RESULTS FOR SCANNED DOCUMENTS.

We further evaluated the effectiveness of our synthetic data generation framework by applying the proposed method to scanned document images. Due to the absence of annotated fraud patterns in existing scanned datasets such as MIDV-2020/SIDTD, we generated our target domain for scanned fraud/non-fraud documents using IDSPACE by randomly selecting control parameters, such as noise level, brightness, contrast, sharpness, and subtle blurring, as illustrated in Tab. 1. For instance, the noise level and contrast were sampled from the range $[0.5, 1.5]$, while the shadow shift was randomly chosen from the range $[-3, 3]$. The ranges were selected based on visual inspection and alignment with scanned (non-fraud) documents from the MIDV dataset to ensure visual realism. These control parameters, once selected, are used to generate all documents of each type. Most of the user-defined metadata (also illustrated in Tab. 1 is randomly sampled for each document generation so that each generated document may differ in these aspects (e.g., different position and orientation).

Using this approach, we created a small dataset consisting of 100 non-fraud images and 100 *inpaint-and-rewrite* fraud images for each country.

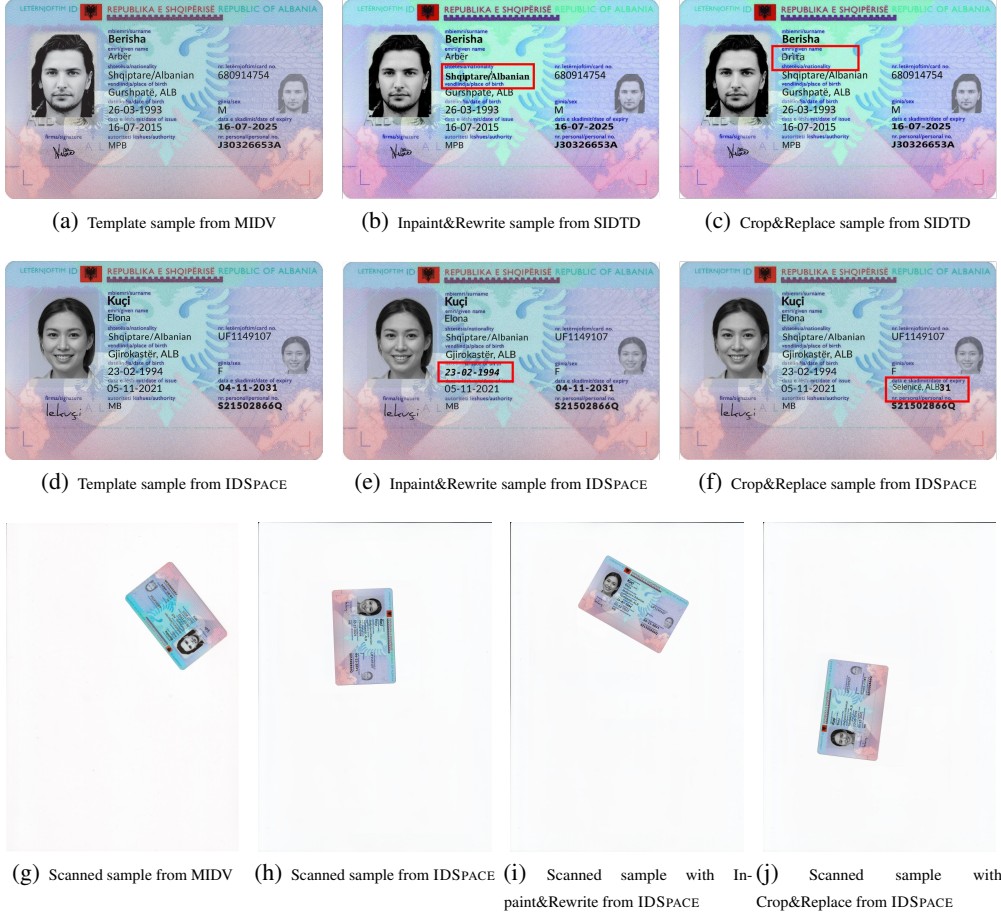

(a) Template sample from MIDV   (b) Inpaint&Rewrite sample from SIDTD   (c) Crop&Replace sample from SIDTD

(d) Template sample from IDSPACE   (e) Inpaint&Rewrite sample from IDSPACE   (f) Crop&Replace sample from IDSPACE

(g) Scanned sample from MIDV   (h) Scanned sample from IDSPACE   (i) Scanned sample with Inpaint&Rewrite from IDSPACE   (j) Scanned sample with Crop&Replace from IDSPACE

Figure 4: Examples from multiple datasets containing Albanian ID card images.

We followed the same methodology as with the template images: training models on the generated dataset and performing Bayesian Optimization (BO) with and without model guidance to search over the predefined parameter set. We then evaluated the consistency scores using different combinations of guiding models for the BO process with scanned documents in Albania and Finland. The results are presented in Table 6 and Table 7, respectively.

From these results, we observe that the consistency scores of the model-guided BO method are generally higher than those obtained on template images. The explanation is that the template region in scanned images typically occupies only around $20\%$ of the entire image. As a result, the SSIM-only objective is not sufficiently sensitive to capture subtle manipulations in these regions, allowing the model-guided approach to dominate in optimizing the objective function.

## A.4    PARAMETER VARIATION ACROSS AND ALONG BAYESIAN OPTIMIZATION PROCESS

**Variation across Optimization Processes.** To evaluate the stability of Bayesian Optimization (BO) across repeated runs for each model, we computed the mean pairwise distance between the best parameter vectors obtained in different runs. Let $\mathbf{x}^{(i)}$ and $\mathbf{x}^{(j)}$ denote two parameter vectors, each containing $p$ normalized generation parameters (i.e., parameters used for generating the documents). Their Euclidean distance is defined as

$$d(\mathbf{x}^{(i)}, \mathbf{x}^{(j)}) = \sqrt{\sum_{k=1}^{p} \left(x_k^{(i)} - x_k^{(j)}\right)^2}.$$

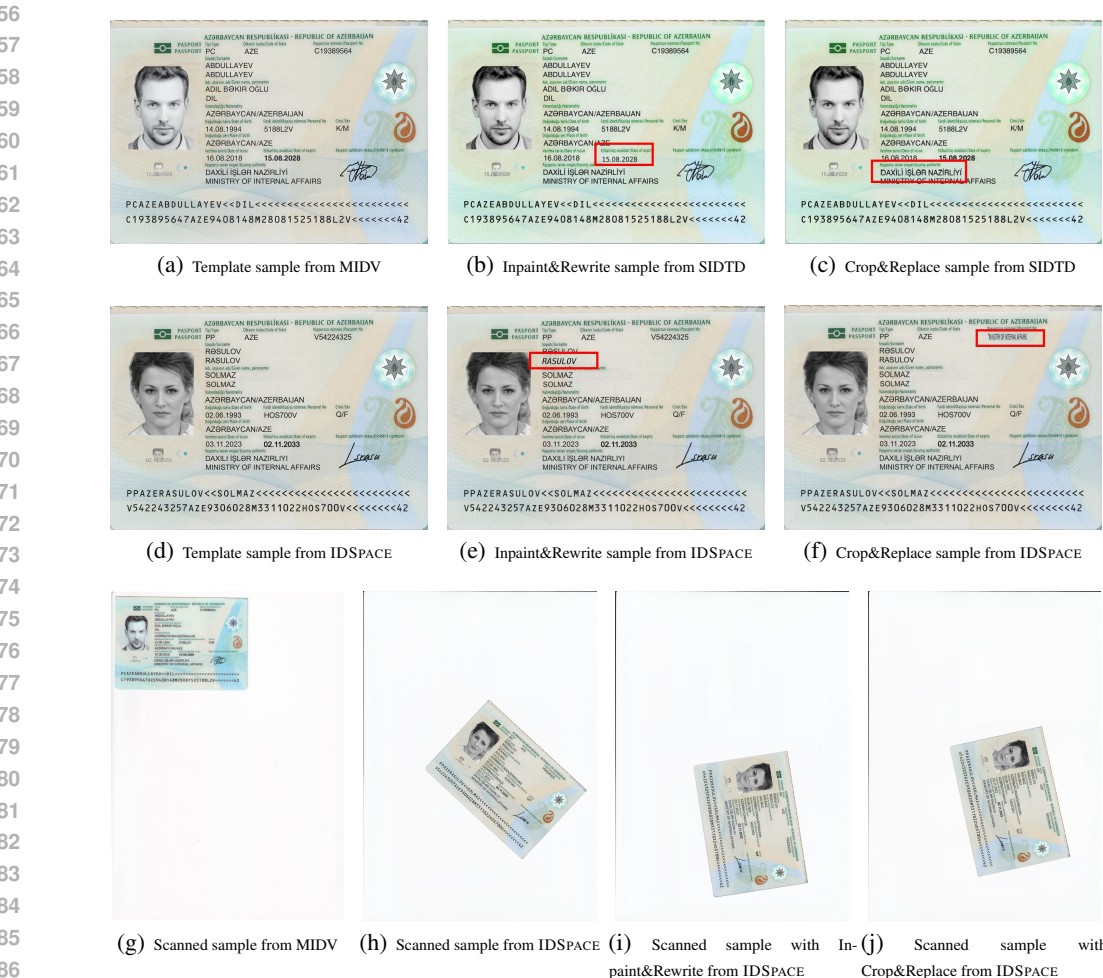

(a) Template sample from MIDV    (b) Inpaint&Rewrite sample from SIDTD    (c) Crop&Replace sample from SIDTD

(d) Template sample from IDSPACE    (e) Inpaint&Rewrite sample from IDSPACE    (f) Crop&Replace sample from IDSPACE

(g) Scanned sample from MIDV    (h) Scanned sample from IDSPACE    (i) Scanned sample with Inpaint&Rewrite from IDSPACE    (j) Scanned sample with Crop&Replace from IDSPACE

Figure 5: Examples from multiple datasets containing Azerbaijani passport images.

For each model, all pairwise distances were calculated among its $n$ runs, and the mean pairwise distance $\bar{d}$ was taken as an indicator of the variance of the parameters along the optimization process. A lower $\bar{d}$ value indicates that BO repeatedly converged to similar parameter configurations, implying a well-defined optimum, whereas a higher $\bar{d}$ suggests multiple local optima or instability in the optimization landscape.

**Variation along Optimization Processes.** To identify which generation parameters contributed most to this optimization variability, we further computed the standard deviation of each parameter across all BO runs for a given model. For a parameter $k$, its variability was quantified as

$$\sigma_k = \sqrt{\frac{1}{n-1} \sum_{i=1}^{n} \left( x_k^{(i)} - \bar{x}_k \right)^2},$$

where $x_k^{(i)}$ is the normalized value of parameter $k$ in the $i$-th run, and $\bar{x}_k$ is its mean across runs. The parameters with the highest $\sigma_k$ values were reported as the most unstable or influential parameters. This analysis highlights which aspects of the generation process BO adjusted most aggressively, thereby providing insight into which parameters dominate the optimization dynamics and sensitivity landscape.

Tab. **??** shows the $d$ for each varying model combination and important parameters with the top $\sigma_k$ values. Three consistent trends emerge from this variation analysis.

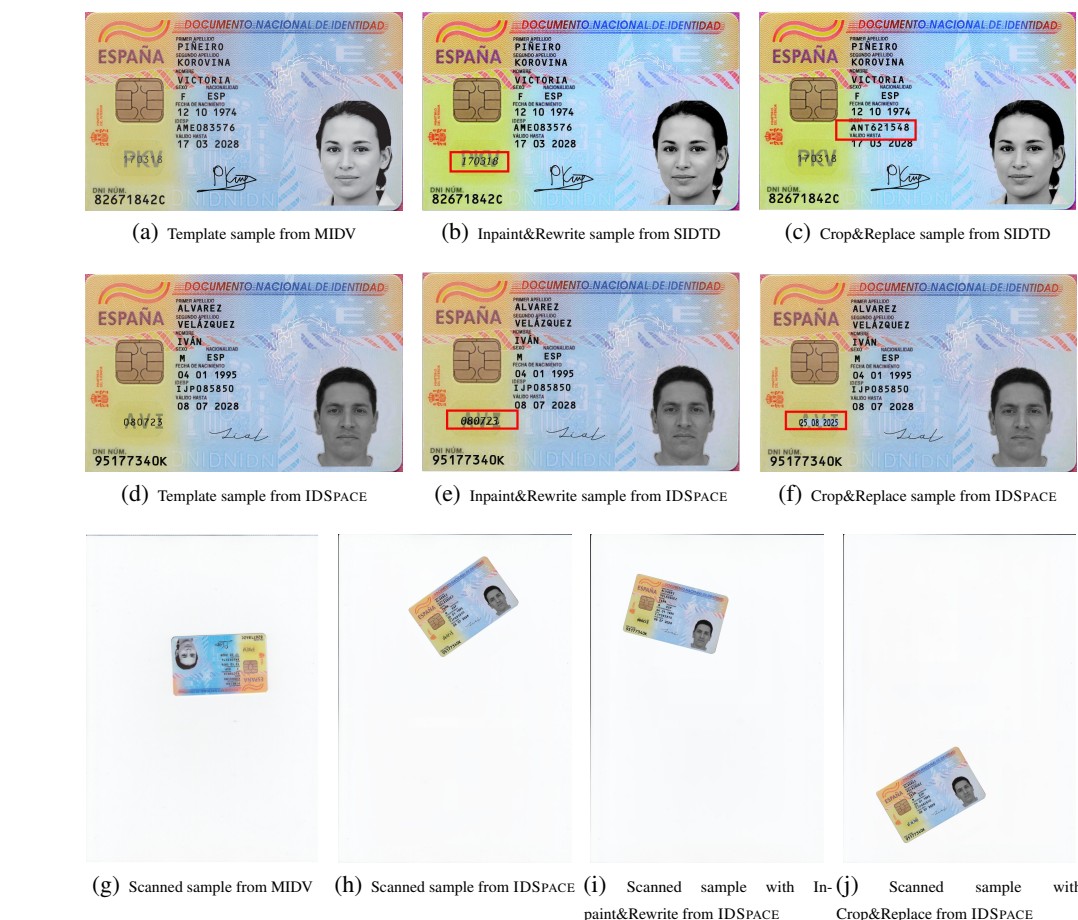

Figure 6: Examples from multiple datasets containing Spanish ID card images.

• Models optimized jointly usually (e.g., two- or three-model combinations) exhibit less variance than their single-model counterparts. This indicates that multi-model BO acts as a regularizer: because the joint objective must satisfy multiple architectures simultaneously, the search converges toward parameter regions that are robust across models, resulting in more stable and predictable behavior.

• The centroid distances for pairwise combinations are typically between the distances of the two constituent models. In other words, the optimal parameters of an $A\_B$ model lie approximately between the optima of $A$ and $B$. This confirms that BO is performing a *negotiated compromise*: no single model dominates the combined run, and the optimizer selects parameter settings beneficial to both. This pattern also holds for the three-model combination, whose centroid lies in the convex region spanned by the individual models.

• It tends to exhibit higher variance along the parameters for which the objective provides little discriminative signal. In such directions, the loss landscape is effectively flat, enabling BO to select widely varying values across runs without impacting the combined SSIM and consistency objective. This effect is most pronounced for `xc`, `yc`, and `zc`, which encode RGB color channels; because SSIM is largely insensitive to global color shifts, BO receives no signal to favor a specific color configuration, leading to substantial across-run variability. The `stroke_w` parameter (representing stroke width) also exhibits high variance, but for a different reason: its search space is binary (`0` or `1`), so any mixing between the two discrete modes produces a large empirical variance even when BO is behaving consistently. Therefore, high variance in `stroke_width` may reflect the discrete nature of the parameter rather than instability in the optimization process.

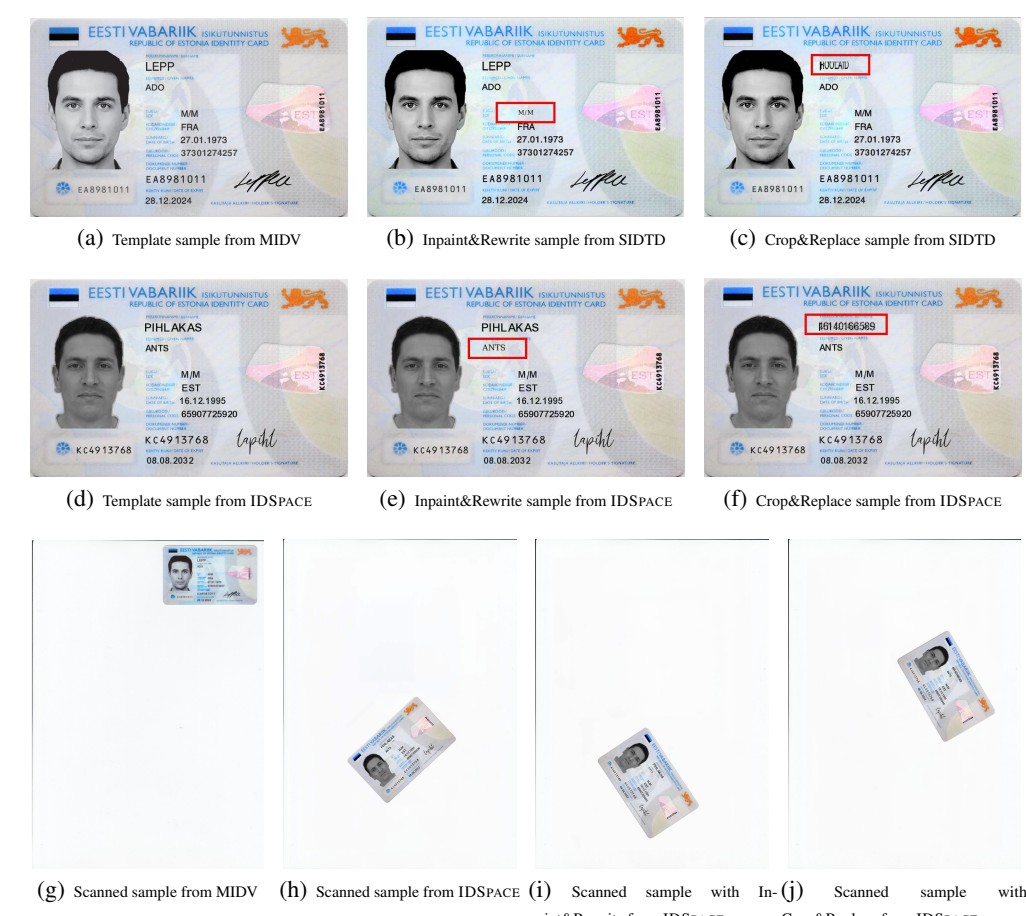

(a) Template sample from MIDV    (b) Inpaint&Rewrite sample from SIDTD    (c) Crop&Replace sample from SIDTD

(d) Template sample from IDSPACE    (e) Inpaint&Rewrite sample from IDSPACE    (f) Crop&Replace sample from IDSPACE

(g) Scanned sample from MIDV    (h) Scanned sample from IDSPACE    (i) Scanned sample with Inpaint&Rewrite from IDSPACE    (j) Scanned sample with Crop&Replace from IDSPACE

Figure 7: Examples from multiple datasets containing Estonian ID card images.

## A.5 DIFFERENCES ACROSS PARAMETERS CHOSEN BY USING DIFFERENT GUIDING MODELS

To compare the overall parameter preferences of using different guiding models, we computed the inter-model centroid distances. For each model $m$, the centroid $\boldsymbol{\mu}^{(m)}$ represents the mean of its normalized optimal-parameter vectors across all BO runs:

$$\boldsymbol{\mu}^{(m)} = \frac{1}{n_m} \sum_{i=1}^{n_m} \mathbf{x}_i^{(m)},$$

where $\mathbf{x}_i^{(m)}$ denotes the $i$-th run's parameter vector. The pairwise distance between two models $m_1$ and $m_2$ was then defined as the Euclidean distance between their parameter centroids:

$$D(m_1, m_2) = \sqrt{\sum_{k=1}^{p} \left( \mu_k^{(m_1)} - \mu_k^{(m_2)} \right)^2}.$$

where $p$ is the number of normalized parameters. A smaller $D(m_1, m_2)$ indicates that the models favor similar generation parameters, while larger distances suggest distinct data-generation sensitivities. This centroid-based measure provides a compact summary of inter-model differences in the optimization landscape.

The pair-wise optimal-parameter-distance matrix is shown in Table **??**. A consistent pattern is observed for the combined models: each combined model lies closer to its constituent models than the constituents lie to each other. For example, the distance between ResNet50 and VGG16

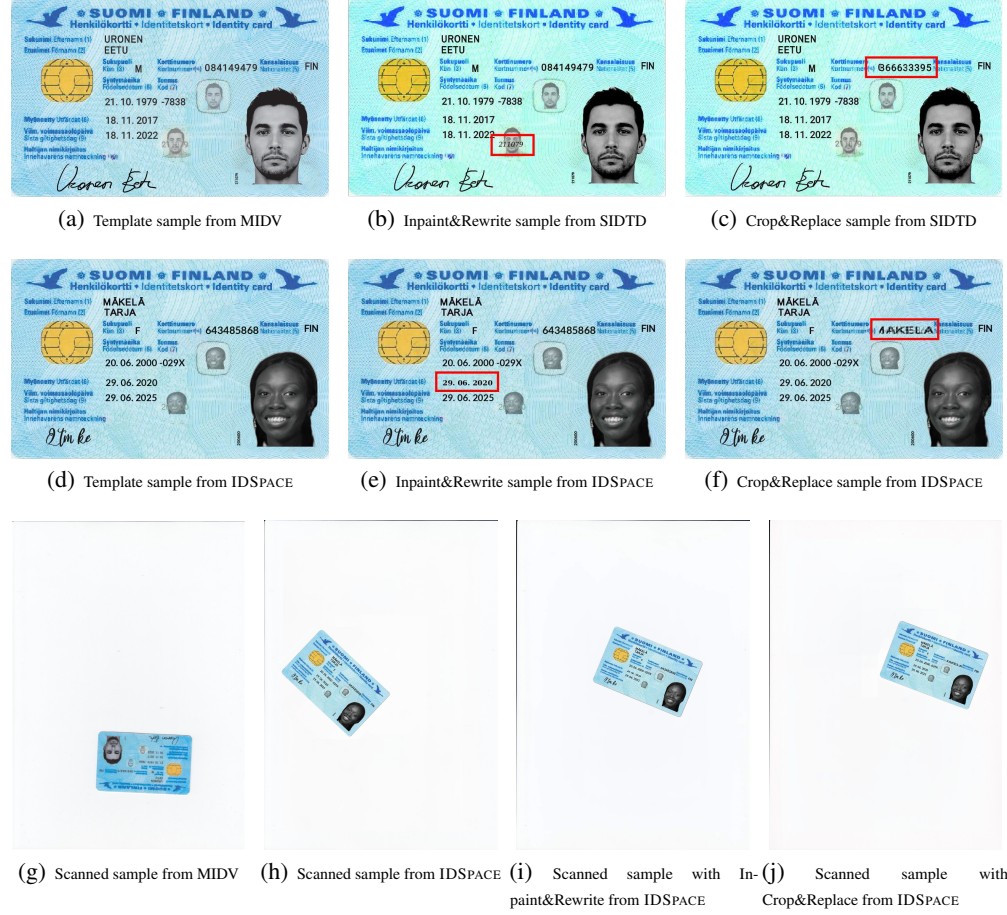

(a) Template sample from MIDV  (b) Inpaint&Rewrite sample from SIDTD  (c) Crop&Replace sample from SIDTD

(d) Template sample from IDSPACE  (e) Inpaint&Rewrite sample from IDSPACE  (f) Crop&Replace sample from IDSPACE

(g) Scanned sample from MIDV  (h) Scanned sample from IDSPACE  (i) Scanned sample with Inpaint&Rewrite from IDSPACE  (j) Scanned sample with Crop&Replace from IDSPACE

Figure 8: Examples from multiple datasets containing Finnish ID card images.

is large ($D = 0.85$), yet the two-model combination ResNet50+VGG16 is substantially closer to both ($D = 0.70$ from ResNet50 and $D = 0.43$ from VGG16). A similar effect holds for the ViT-Large+ResNet50 pair: although ResNet50 and ViT-Large are separated by $D = 0.49$, their combined model lies closer to each component ($D = 0.40$ to ResNet50 and $D = 0.37$ to ViT-Large). The same pattern extends to the three-model combination ViT-Large+ResNet50+VGG16, which is closer to each constituent model ($D = 0.41$, $0.44$, and $0.61$) than some of the constituent pairs are to each other (e.g., R50–VGG16 at $D = 0.85$ and VGG16–ViT-Large at $D = 0.66$).

The smallest distance in the table, $D = 0.21$, occurs for VGG16 and ViT-Large+VGG16, showing that this particular hybrid aligns especially closely with one of its components. In contrast, larger distances (e.g., $D = 0.85$ for ResNet50–VGG16) indicate model-specific parameter preferences.

This consistent reduction in distance indicates that joint optimization pulls the solution toward a shared region of parameter space that simultaneously satisfies all participating models. In other words, the combined-model centroid is always an *interpolated* solution lying between its components rather than a degenerate solution that drifts toward a single model or moves outside the span of the individual optima. This provides strong evidence that multi-model BO converges to a compromise configuration that reduces disagreement between models in the generation-parameter space.

## A.6 SCALING TO NEW DOCUMENT TYPES.

We released documents from only ten European countries because each of these document types has around 100 high-quality identity documents for non-fraud and fraud classes, respectively, enabling us to train and drive our model-guided approach to generate meaningful synthetic data. These ten types

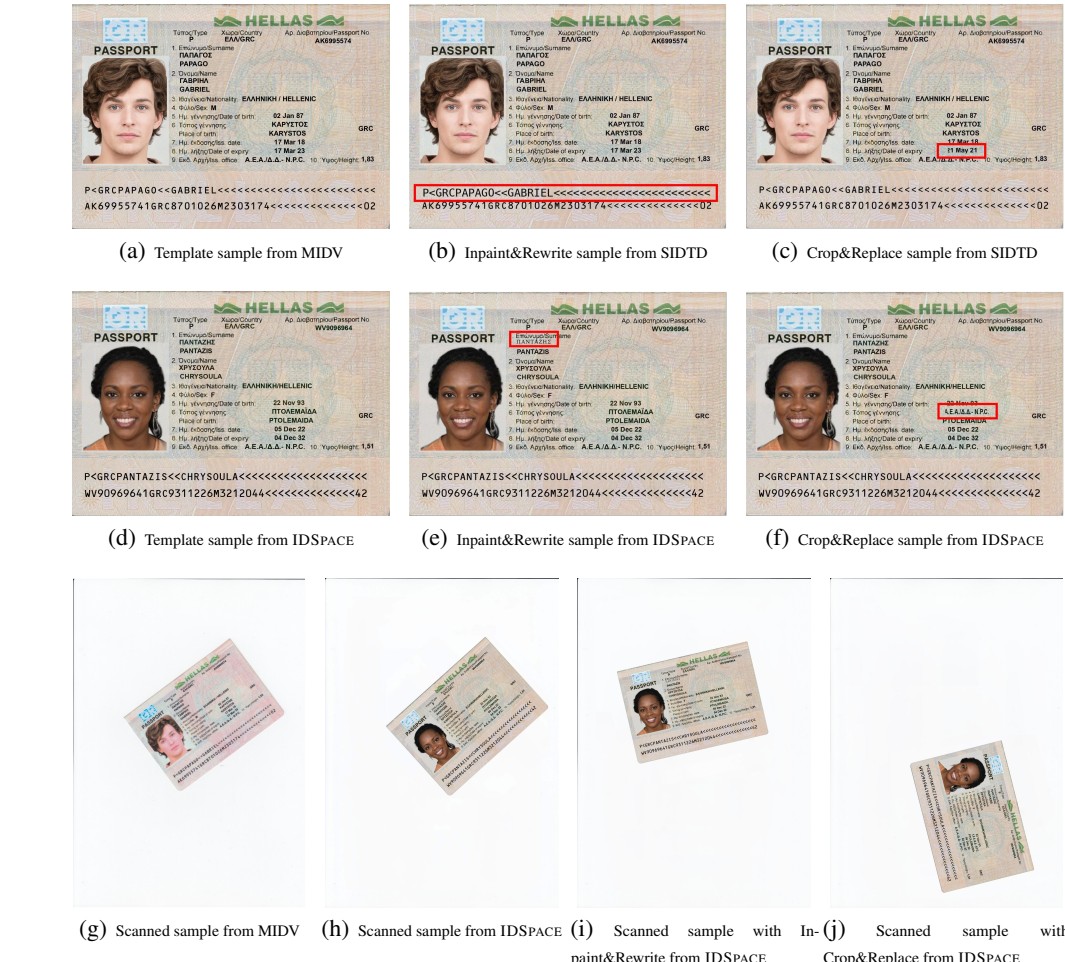

(a) Template sample from MIDV

(b) Inpaint&Rewrite sample from SIDTD

(c) Crop&Replace sample from SIDTD

(d) Template sample from IDSPACE

(e) Inpaint&Rewrite sample from IDSPACE

(f) Crop&Replace sample from IDSPACE

(g) Scanned sample from MIDV

(h) Scanned sample from IDSPACE

(i) Scanned sample with Inpaint&Rewrite from IDSPACE

(j) Scanned sample with Crop&Replace from IDSPACE

Figure 9: Examples from multiple datasets containing Greek passport images.

of documents, including five types of passports and five types of ID cards, encompass most of the security features and schema elements found in general identity documents. They also cover different languages.

To demonstrate that our technique could apply to other types, we used the synthetic West Virginia driver's licenses dataset from prior work Xie et al. (2024); Guan et al. (2024), which consists of 5,979 synthetic non-fraud documents and 5,979 synthetic fraud documents for each of the fraud patterns.

We trained the EfficientNet-b3, ResNet50, and ViT-Large models using 1000 non-fraud documents and 1000 fraud documents, which are split into training, validation, and testing sets by 5:2:3. Similar to the experimental setting used for Table 2 in our submitted paper, we used 20 non-fraud documents and 20 fraud documents for Bayesian Optimization search with and without guiding models for finetuning the hyper-parameters.

The model consistency evaluation results (similar to Tab. 2), obtained on the testing sets (including 300 non-fraud documents and 300 fraud documents), are illustrated in Tab. 10 and Tab. 11, highlighting the good generalization capability of our IDSPACE synthetic data generation approach proposed in this work.

Adding a new document template does require an engineering pass to identify field positions and mapping logic. Still, we emphasize that each new template requires a one-time specification of field mappings; however, the effort is modest and is amortized by the scalable generation. This cost is orders of magnitude smaller than collecting and annotating new real data.

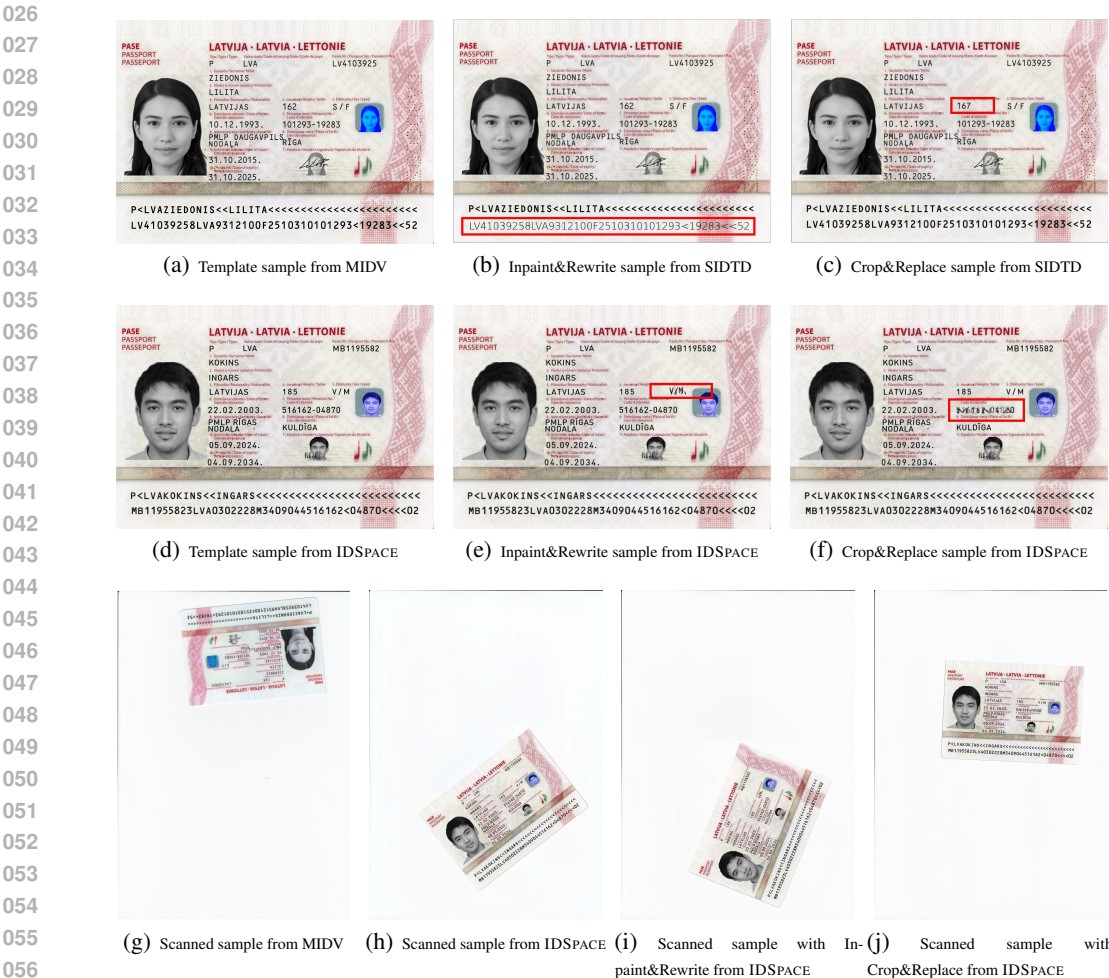

(a) Template sample from MIDV    (b) Inpaint&Rewrite sample from SIDTD    (c) Crop&Replace sample from SIDTD

(d) Template sample from IDSPACE    (e) Inpaint&Rewrite sample from IDSPACE    (f) Crop&Replace sample from IDSPACE

(g) Scanned sample from MIDV    (h) Scanned sample from IDSPACE    (i) Scanned sample with Inpaint&Rewrite from IDSPACE    (j) Scanned sample with Crop&Replace from IDSPACE

Figure 10: Examples from multiple datasets containing Latvian passport images.

Tab. 12 illustrates a detailed breakdown of the time spent preparing a new template for the above experiment on the West Virginia Driver's License dataset by a Ph.D student with one year of experience in the identity document design domain:

The automatic processing times are illustrated in Tab. 13, which are tested on an Ubuntu Linux server equipped with 48 CPU cores (Intel Xeon Silver 4310, 2.10 GHz), 125 GB of RAM, and two NVIDIA A10 GPUs (24GB VRAM each).

### A.7 STYLEGAN RESULTS.

To evaluate the identity generation capability of GANs, we fine-tuned a pre-trained StyleGAN model using 40 Albanian identity documents in the MIDV/SIDTD dataset. We adopted the official StyleGAN3 training configuration with the following parameters: `cfg=stylegan2`, `snap=10`, `mirror=1`, `batch=32`, and `gamma=8.2`. Training was conducted using the pretrained model *stylegan2-ffhq-1024x1024.pkl*. Since StyleGAN3 requires square inputs, we resized the original image dimensions from $(2167, 1360)$ to $(1024, 1024)$, preserving the aspect ratio as much as possible. Model performance was assessed using the FID50K_full metric (Fréchet Inception Distance) against the full dataset. The best FID score of 57.52 was achieved using the pretrained model, and the corresponding checkpoint was selected for evaluation.

Figure 14 presents example output. As illustrated, the model struggles to generate semantically meaningful or identity-consistent images mainly due to the limited size of the training dataset. GANs

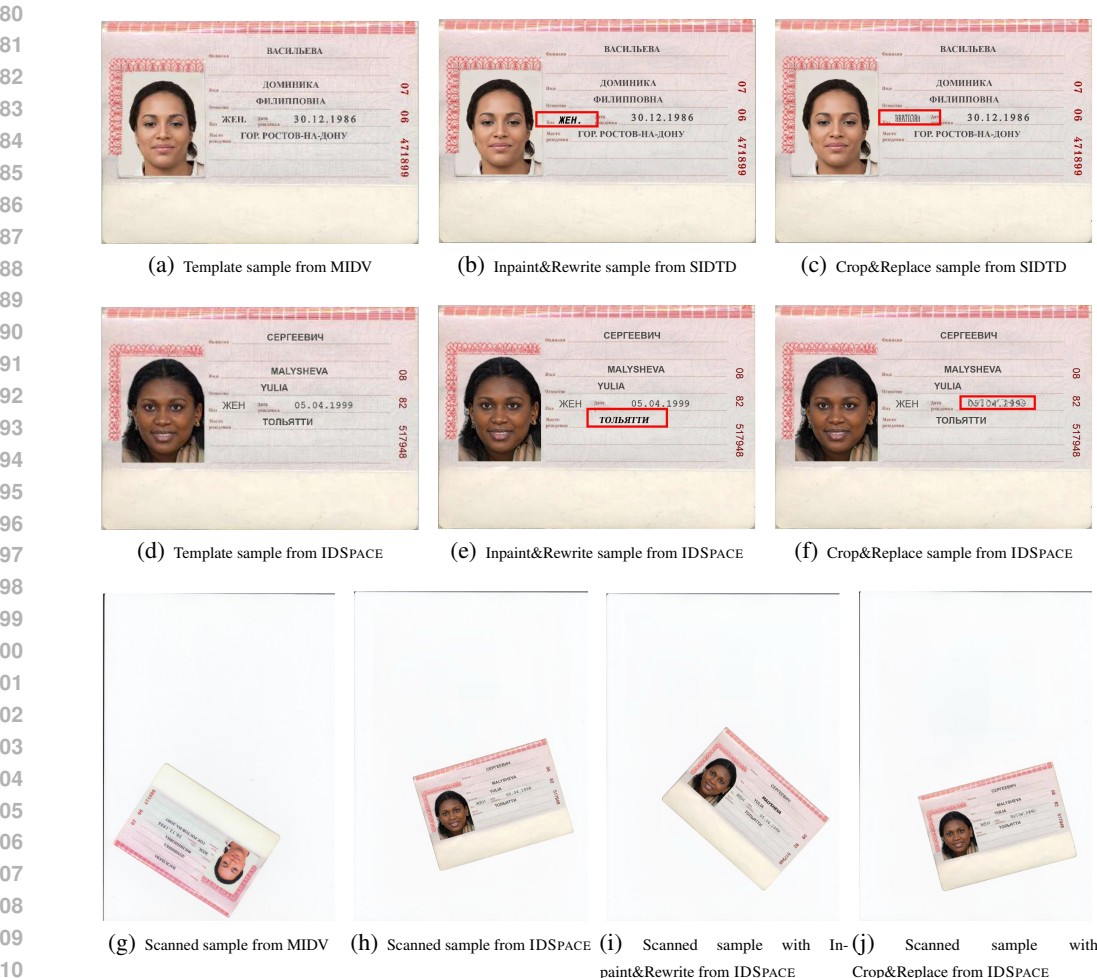

(a) Template sample from MIDV

(b) Inpaint&Rewrite sample from SIDTD

(c) Crop&Replace sample from SIDTD

(d) Template sample from IDSPACE

(e) Inpaint&Rewrite sample from IDSPACE

(f) Crop&Replace sample from IDSPACE

(g) Scanned sample from MIDV

(h) Scanned sample from IDSPACE

(i) Scanned sample with Inpaint&Rewrite from IDSPACE

(j) Scanned sample with Crop&Replace from IDSPACE

Figure 11: Examples from multiple datasets containing Russian passport images.

generally require hundreds to thousands of high-quality images to learn robust and high-fidelity representations.

## A.8 DIFFUSION RESULTS.

To evaluate the capacity of diffusion models to generate realistic identity documents, we fine-tuned a pre-trained Stable Diffusion model (`stable-diffusion-v1-5`) using a subset of the MIDV/SIDTD dataset consisting of **40 Albanian ID cards**. We adopted a LoRA-based fine-tuning setup and resized the original images from $2167 \times 1360$ to $1024 \times 768$ to match the model's maximum resolution while approximately preserving aspect ratio. The model was trained for **5000 epochs**, and we saved checkpoints every 500 epochs. We further experimented with LoRA ranks of 64 and 128.

Because the Stable Diffusion text encoder limits prompts to **77 CLIP tokens**, we compressed each ground-truth OCR string into the following prompt:

```
Albanianid template; portrait female; Cobaj Elona; 26091957;
Mgull, ALB; MB; F75926997V; 499949517; 19022018; 19022028.
```

For each checkpoint, we generated five samples using this fixed prompt.

Figure 15 shows the most visually plausible output, obtained at epoch 1500 with LoRA rank 128. Even in this best case, the model fails to faithfully reproduce the structural layout and fine-grained security features of Albanian ID cards.

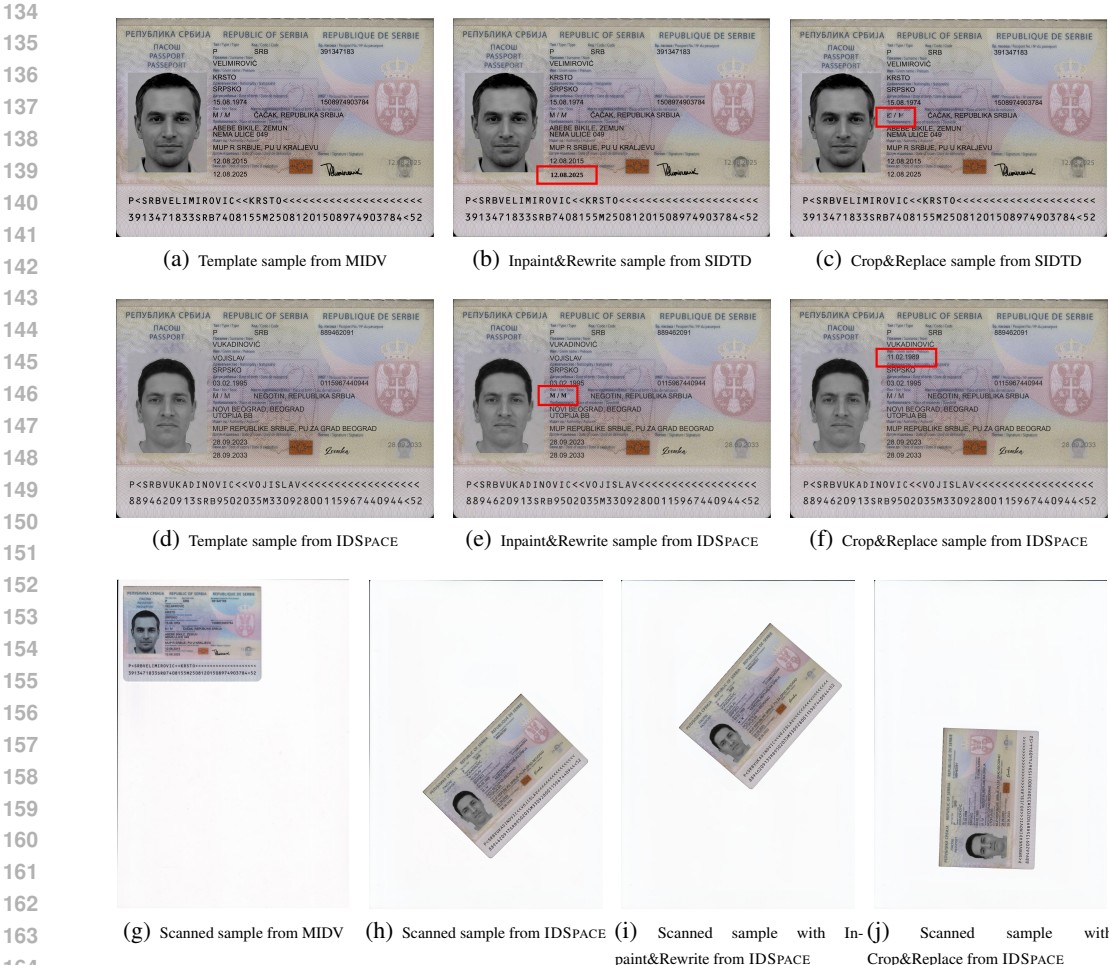

(a) Template sample from MIDV

(b) Inpaint&Rewrite sample from SIDTD

(c) Crop&Replace sample from SIDTD

(d) Template sample from IDSPACE

(e) Inpaint&Rewrite sample from IDSPACE

(f) Crop&Replace sample from IDSPACE

(g) Scanned sample from MIDV

(h) Scanned sample from IDSPACE

(i) Scanned sample with In-paint&Rewrite from IDSPACE

(j) Scanned sample with Crop&Replace from IDSPACE

Figure 12: Examples from multiple datasets containing Serbian passport images.

**Why does the model struggle?** Diffusion models such as Stable Diffusion are optimized for natural images and semantically grounded objects, not for generating *highly structured, low-entropy, layout-constrained* document formats. ID cards contain dense microtext, rigid field alignment, intricate security patterns, and strict template geometry—none of which are well represented in the model's pre-training distribution. Given that only 40 examples are available for fine-tuning, the adaptation signal is insufficient to counteract the strong natural-image prior learned during large-scale training. Consequently, the model hallucinates textures, distorts global layout, and produces blurred textual regions instead of replicating the precise template structure of authentic identity documents.

## A.9 GENERATIVE RESULTS.

We also experimented with using GPT-4o and GPT-image-1 for ID image generation. The following prompt was provided for both of the models: *"Using the provided sample as a reference, generate a realistic-looking ID card that closely mimics the layout, design, and visual elements. Replace all personal information (name, date of birth, ID number, place of birth, etc.) with clearly fictional data. Ensure that all formatting, fonts, and security features (such as watermarks, holograms, and layout positioning) remain as similar to the original as possible."*

Figure 16 presents four images: the original sample (a), the image generated by GPT-4o (b), the image generated by IDSPACE (c), and the image generated by GPT-Image-1 (d). As shown, the image generated by GPT-4o replicates certain background elements; however, it fails to fully preserve the original template structure. While some personal details were modified, others—such as the expiry

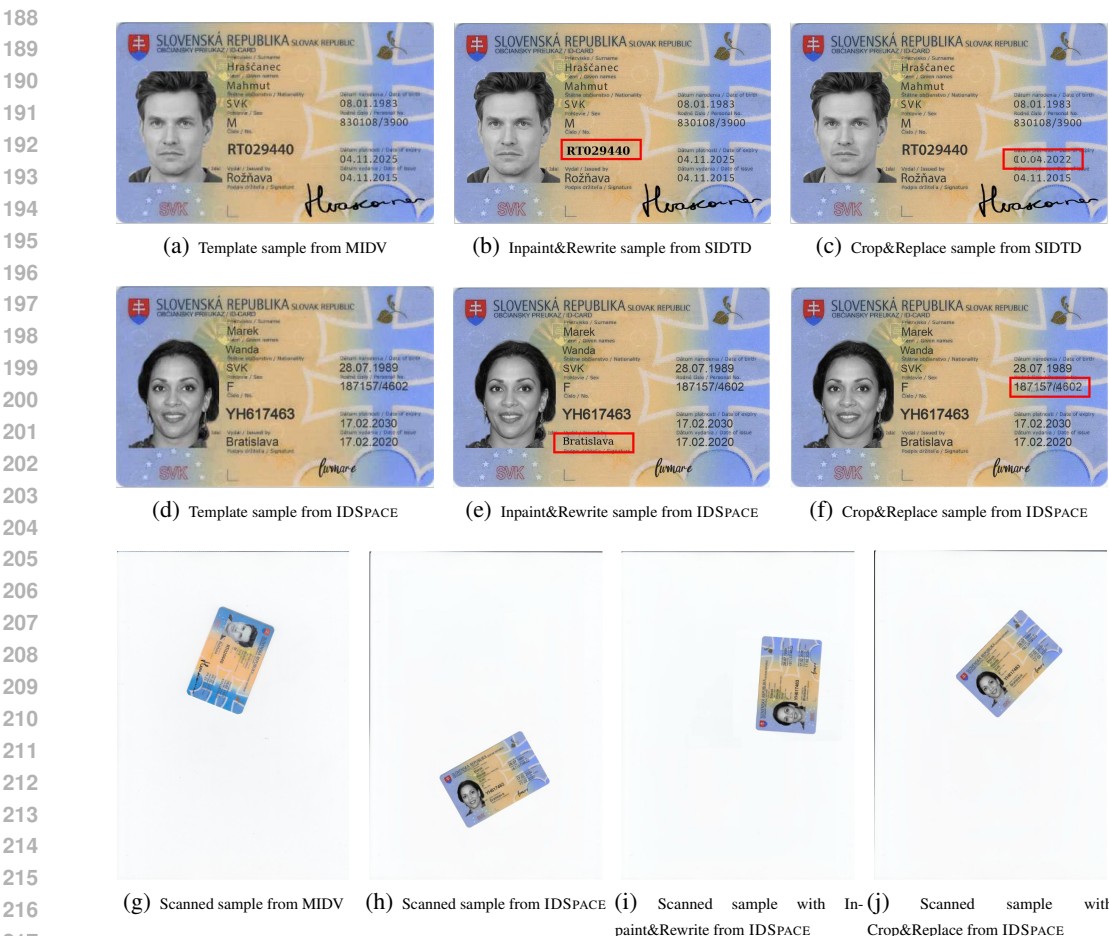

(a) Template sample from MIDV

(b) Inpaint&Rewrite sample from SIDTD

(c) Crop&Replace sample from SIDTD

(d) Template sample from IDSPACE

(e) Inpaint&Rewrite sample from IDSPACE

(f) Crop&Replace sample from IDSPACE

(g) Scanned sample from MIDV

(h) Scanned sample from IDSPACE

(i) Scanned sample with Inpaint&Rewrite from IDSPACE

(j) Scanned sample with Crop&Replace from IDSPACE

Figure 13: Examples from multiple datasets containing Slovakian ID card images.

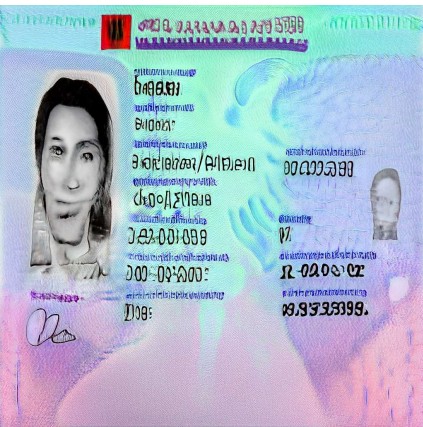

Figure 14: Images generated by StyleGAN3 using 40 identity samples from the MIDV/SIDTD dataset.

date—remained unchanged. Additionally, the layout was inconsistently altered, affecting fields such as the issuing authority, personal number, and date of issue.

The image generated by GPT-Image-1 demonstrates a clear improvement over GPT-4o by successfully modifying all personal information as instructed. However, it changes the background design of the

Table 5: Model prediction consistency evaluation on different models with best results highlighted in blue (FIN)

| Guiding models | ViT-Large | ResNet50 | Inception-v3 | VGG16 | DenseNet | Average |
|---|---|---|---|---|---|---|
| SSIM-only objective | 0.5143 ± 0.007 | 0.5238 ± 0.040 | 0.5667 ± 0.015 | 0.7714 ± 0.004 | 0.5952 ± 0.050 | 0.5943 ± 0.009 |
| DenseNet | 0.8667 ± 0.002 | 0.8762 ± 0.007 | 0.8571 ± 0.003 | 0.8714 ± 0.000 | 0.9905 ± 0.000 | 0.8924 ± 0.002 |
| DenseNet + Inception-v3 | 0.9000 ± 0.003 | 0.8762 ± 0.003 | 0.8905 ± 0.005 | 0.8667 ± 0.001 | 0.9857 ± 0.002 | 0.9038 ± 0.002 |
| DenseNet + Inception-v3 + ResNet50 | 0.9333 ± 0.001 | 0.8905 ± 0.003 | 0.8714 ± 0.001 | 0.8857 ± 0.000 | 0.9952 ± 0.000 | 0.9152 ± 0.002 |
| DenseNet + Inception-v3 + VGG16 | 0.9190 ± 0.002 | 0.9048 ± 0.001 | 0.9190 ± 0.002 | 0.8905 ± 0.000 | 1.0000 ± 0.000 | 0.9267 ± 0.001 |
| DenseNet + Inception-v3 + ViT-Large | 0.9143 ± 0.002 | 0.9048 ± 0.002 | 0.9000 ± 0.002 | 0.8810 ± 0.001 | 0.9667 ± 0.002 | 0.9133 ± 0.001 |
| DenseNet + ResNet50 | 0.7905 ± 0.002 | 0.9190 ± 0.002 | 0.8238 ± 0.003 | 0.8810 ± 0.000 | 0.9952 ± 0.000 | 0.8819 ± 0.005 |
| DenseNet + ResNet50 + VGG16 | 0.8952 ± 0.004 | 0.8952 ± 0.001 | 0.8476 ± 0.005 | 0.8762 ± 0.000 | 0.9524 ± 0.003 | 0.8933 ± 0.001 |
| DenseNet + ResNet50 + ViT-Large | 0.8476 ± 0.006 | 0.8905 ± 0.004 | 0.8095 ± 0.001 | 0.8905 ± 0.001 | 0.9190 ± 0.003 | 0.8714 ± 0.001 |
| DenseNet + VGG16 | 0.8381 ± 0.002 | 0.8714 ± 0.002 | 0.8000 ± 0.004 | 0.8619 ± 0.001 | 0.9714 ± 0.001 | 0.8686 ± 0.003 |
| DenseNet + VGG16 + ViT-Large | 0.9667 ± 0.000 | 0.9238 ± 0.000 | 0.9143 ± 0.001 | 0.8952 ± 0.000 | 0.9143 ± 0.000 | 0.9229 ± 0.001 |
| DenseNet + ViT-Large | 0.8762 ± 0.003 | 0.8952 ± 0.002 | 0.7952 ± 0.008 | 0.8476 ± 0.001 | 0.9571 ± 0.001 | 0.8743 ± 0.003 |
| Inception-v3 | 0.9048 ± 0.002 | 0.8619 ± 0.004 | 0.8619 ± 0.006 | 0.8667 ± 0.000 | 0.9810 ± 0.000 | 0.8952 ± 0.002 |
| Inception-v3 + ResNet50 | 0.9571 ± 0.000 | 0.9095 ± 0.000 | 0.9571 ± 0.001 | 0.8905 ± 0.000 | 1.0000 ± 0.000 | 0.9429 ± 0.002 |
| Inception-v3 + ResNet50 + VGG16 | 0.8095 ± 0.006 | 0.9190 ± 0.001 | 0.8238 ± 0.002 | 0.8714 ± 0.000 | 0.8762 ± 0.008 | 0.8600 ± 0.002 |
| Inception-v3 + ResNet50 + ViT-Large | 0.9000 ± 0.004 | 0.8714 ± 0.001 | 0.8810 ± 0.001 | 0.8667 ± 0.001 | 0.9762 ± 0.001 | 0.8990 ± 0.002 |
| Inception-v3 + VGG16 | 0.9571 ± 0.000 | 0.9667 ± 0.000 | 0.9381 ± 0.000 | 0.9095 ± 0.000 | 0.8952 ± 0.000 | 0.9333 ± 0.001 |
| Inception-v3 + VGG16 + ViT-Large | 0.9286 ± 0.003 | 0.9190 ± 0.001 | 0.9238 ± 0.002 | 0.8952 ± 0.000 | 0.9238 ± 0.003 | 0.9181 ± 0.000 |
| Inception-v3 + ViT-Large | 0.9333 ± 0.001 | 0.9238 ± 0.000 | 0.8571 ± 0.005 | 0.8905 ± 0.000 | 0.9333 ± 0.001 | 0.9076 ± 0.001 |
| ResNet50 | 0.8190 ± 0.002 | 0.8571 ± 0.001 | 0.7048 ± 0.004 | 0.8476 ± 0.000 | 0.8429 ± 0.000 | 0.8143 ± 0.003 |
| ResNet50 + VGG16 | 0.8667 ± 0.001 | 0.8905 ± 0.001 | 0.8286 ± 0.001 | 0.8762 ± 0.000 | 0.9667 ± 0.001 | 0.8857 ± 0.002 |
| ResNet50 + VGG16 + ViT-Large | 0.9143 ± 0.002 | 0.8619 ± 0.003 | 0.8857 ± 0.005 | 0.8857 ± 0.001 | 0.9857 ± 0.002 | 0.9067 ± 0.002 |
| ResNet50 + ViT-Large | 0.8905 ± 0.001 | 0.9429 ± 0.001 | 0.8952 ± 0.002 | 0.8857 ± 0.001 | 0.8905 ± 0.003 | 0.9010 ± 0.000 |
| VGG16 | 0.8000 ± 0.010 | 0.8810 ± 0.003 | 0.7476 ± 0.007 | 0.8571 ± 0.001 | 0.8238 ± 0.030 | 0.8219 ± 0.002 |
| VGG16 + ViT-Large | 0.9524 ± 0.000 | 0.9238 ± 0.001 | 0.9048 ± 0.003 | 0.8905 ± 0.000 | 0.9571 ± 0.004 | 0.9257 ± 0.001 |
| ViT-Large | 0.9524 ± 0.000 | 0.9429 ± 0.000 | 0.9333 ± 0.000 | 0.9048 ± 0.000 | 0.9000 ± 0.000 | 0.9267 ± 0.000 |

Table 6: Model prediction consistency evaluation for scanned documents on different models with best results highlighted in blue (ALB)

| Guiding models | ViT-Large | ResNet50 | Inception-v3 | VGG16 | DenseNet | Average |
|---|---|---|---|---|---|---|
| SSIM-only objective | 0.4933 ± 0.000 | 0.4933 ± 0.000 | 0.4933 ± 0.000 | 0.4933 ± 0.000 | 0.5733 ± 0.000 | 0.5093 ± 0.001 |
| DenseNet | 1.0000 ± 0.000 | 0.9778 ± 0.000 | 0.9867 ± 0.000 | 1.0000 ± 0.000 | 0.9778 ± 0.000 | 0.9884 ± 0.000 |
| DenseNet + Inception-v3 | 1.0000 ± 0.000 | 0.9733 ± 0.000 | 0.9822 ± 0.000 | 0.9689 ± 0.002 | 0.9600 ± 0.000 | 0.9769 ± 0.000 |
| DenseNet + Inception-v3 + ResNet50 | 1.0000 ± 0.000 | 0.9733 ± 0.000 | 0.9867 ± 0.000 | 1.0000 ± 0.000 | 0.9556 ± 0.001 | 0.9831 ± 0.000 |
| DenseNet + Inception-v3 + VGG16 | 1.0000 ± 0.000 | 0.9778 ± 0.000 | 0.9867 ± 0.000 | 1.0000 ± 0.000 | 0.9733 ± 0.000 | 0.9876 ± 0.000 |
| DenseNet + Inception-v3 + ViT-Large | 1.0000 ± 0.000 | 0.9822 ± 0.000 | 0.9867 ± 0.000 | 1.0000 ± 0.000 | 0.9733 ± 0.000 | 0.9884 ± 0.000 |
| DenseNet + ResNet50 | 1.0000 ± 0.000 | 0.9733 ± 0.000 | 0.9867 ± 0.000 | 1.0000 ± 0.000 | 0.9733 ± 0.000 | 0.9867 ± 0.000 |
| DenseNet + ResNet50 + VGG16 | 1.0000 ± 0.000 | 0.9778 ± 0.000 | 0.9867 ± 0.000 | 1.0000 ± 0.000 | 0.9511 ± 0.000 | 0.9831 ± 0.000 |
| DenseNet + ResNet50 + ViT-Large | 1.0000 ± 0.000 | 0.9778 ± 0.000 | 0.9867 ± 0.000 | 1.0000 ± 0.000 | 0.9689 ± 0.000 | 0.9867 ± 0.000 |
| DenseNet + VGG16 | 1.0000 ± 0.000 | 0.9911 ± 0.000 | 0.9867 ± 0.000 | 1.0000 ± 0.000 | 0.9689 ± 0.000 | 0.9893 ± 0.000 |
| DenseNet + VGG16 + ViT-Large | 1.0000 ± 0.000 | 0.9778 ± 0.000 | 0.9867 ± 0.000 | 1.0000 ± 0.000 | 0.9778 ± 0.000 | 0.9884 ± 0.000 |
| DenseNet + ViT-Large | 1.0000 ± 0.000 | 0.9822 ± 0.000 | 0.9867 ± 0.000 | 1.0000 ± 0.000 | 0.9644 ± 0.000 | 0.9867 ± 0.000 |
| Inception-v3 | 1.0000 ± 0.000 | 0.9867 ± 0.000 | 0.9867 ± 0.000 | 1.0000 ± 0.000 | 0.9689 ± 0.000 | 0.9884 ± 0.000 |
| Inception-v3 + ResNet50 | 1.0000 ± 0.000 | 0.9778 ± 0.000 | 0.9867 ± 0.000 | 1.0000 ± 0.000 | 0.9644 ± 0.001 | 0.9858 ± 0.000 |
| Inception-v3 + ResNet50 + VGG16 | 1.0000 ± 0.000 | 0.9733 ± 0.000 | 0.9822 ± 0.000 | 1.0000 ± 0.000 | 0.9689 ± 0.000 | 0.9849 ± 0.000 |
| Inception-v3 + ResNet50 + ViT-Large | 1.0000 ± 0.000 | 0.9956 ± 0.000 | 0.9867 ± 0.000 | 1.0000 ± 0.000 | 0.9644 ± 0.000 | 0.9893 ± 0.000 |
| Inception-v3 + VGG16 | 1.0000 ± 0.000 | 0.9733 ± 0.000 | 0.9867 ± 0.000 | 1.0000 ± 0.000 | 0.9644 ± 0.000 | 0.9849 ± 0.000 |
| Inception-v3 + VGG16 + ViT-Large | 1.0000 ± 0.000 | 0.9778 ± 0.000 | 0.9867 ± 0.000 | 1.0000 ± 0.000 | 0.9689 ± 0.000 | 0.9867 ± 0.000 |
| Inception-v3 + ViT-Large | 1.0000 ± 0.000 | 0.9733 ± 0.000 | 0.9867 ± 0.000 | 1.0000 ± 0.000 | 0.9600 ± 0.000 | 0.9840 ± 0.000 |
| ResNet50 | 1.0000 ± 0.000 | 0.9867 ± 0.000 | 0.9333 ± 0.006 | 0.9689 ± 0.002 | 0.8978 ± 0.006 | 0.9573 ± 0.001 |
| ResNet50 + VGG16 | 1.0000 ± 0.000 | 0.9911 ± 0.000 | 0.9867 ± 0.000 | 1.0000 ± 0.000 | 0.9822 ± 0.000 | 0.9920 ± 0.000 |
| ResNet50 + VGG16 + ViT-Large | 1.0000 ± 0.000 | 0.9822 ± 0.000 | 0.9867 ± 0.000 | 1.0000 ± 0.000 | 0.9511 ± 0.000 | 0.9840 ± 0.000 |
| ResNet50 + ViT-Large | 1.0000 ± 0.000 | 0.9778 ± 0.000 | 0.9867 ± 0.000 | 1.0000 ± 0.000 | 0.9556 ± 0.000 | 0.9840 ± 0.000 |
| VGG16 | 1.0000 ± 0.000 | 0.9778 ± 0.000 | 0.9867 ± 0.000 | 1.0000 ± 0.000 | 0.9867 ± 0.000 | 0.9902 ± 0.000 |
| VGG16 + ViT-Large | 1.0000 ± 0.000 | 0.9733 ± 0.000 | 0.9822 ± 0.000 | 1.0000 ± 0.000 | 0.9822 ± 0.000 | 0.9876 ± 0.000 |
| ViT-Large | 1.0000 ± 0.000 | 0.9733 ± 0.000 | 0.9244 ± 0.006 | 0.9600 ± 0.003 | 0.9200 ± 0.009 | 0.9556 ± 0.001 |

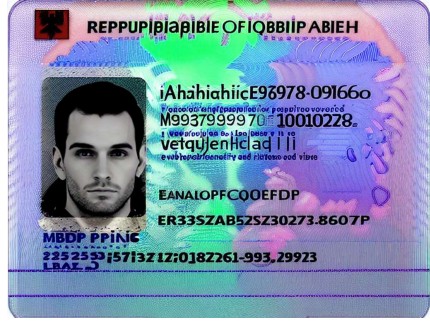

Figure 15: Images generated by diffusion model using 40 identity samples from the MIDV/SIDTD dataset.

ID, resulting in a noticeable difference that can be easily identified by a human observer. Furthermore,

Table 7: Model prediction consistency evaluation for scanned documents on different models with best results highlighted in blue (FIN)

| Guiding models | ViT-Large | ResNet50 | Inception-v3 | VGG16 | DenseNet | Average |
|---|---|---|---|---|---|---|
| SSIM-only objective | 0.4949 ± 0.001 | 0.5505 ± 0.002 | 0.5960 ± 0.013 | 0.5000 ± 0.001 | 0.5505 ± 0.011 | 0.5384 ± 0.001 |
| DenseNet | 0.9848 ± 0.000 | 0.9747 ± 0.000 | 0.9747 ± 0.000 | 0.9646 ± 0.000 | 0.9747 ± 0.000 | 0.9747 ± 0.000 |
| DenseNet + Inception-v3 | 0.9848 ± 0.000 | 0.9798 ± 0.000 | 0.9848 ± 0.000 | 0.9747 ± 0.000 | 1.0000 ± 0.000 | 0.9848 ± 0.000 |
| DenseNet + Inception-v3 + ResNet50 | 0.9848 ± 0.000 | 0.9798 ± 0.000 | 0.9747 ± 0.000 | 0.9697 ± 0.000 | 0.9949 ± 0.000 | 0.9808 ± 0.000 |
| DenseNet + Inception-v3 + VGG16 | 0.9848 ± 0.000 | 0.9646 ± 0.000 | 0.9697 ± 0.000 | 0.9899 ± 0.000 | 1.0000 ± 0.000 | 0.9818 ± 0.000 |
| DenseNet + Inception-v3 + ViT-Large | 0.9848 ± 0.000 | 0.9697 ± 0.000 | 0.9798 ± 0.000 | 0.9697 ± 0.000 | 0.9949 ± 0.000 | 0.9798 ± 0.000 |
| DenseNet + ResNet50 | 0.9848 ± 0.000 | 0.9798 ± 0.000 | 0.9798 ± 0.000 | 0.9747 ± 0.000 | 0.9545 ± 0.001 | 0.9747 ± 0.000 |
| DenseNet + ResNet50 + VGG16 | 0.9848 ± 0.000 | 0.9798 ± 0.000 | 0.9747 ± 0.000 | 0.9747 ± 0.000 | 1.0000 ± 0.000 | 0.9828 ± 0.000 |
| DenseNet + ResNet50 + ViT-Large | 0.9848 ± 0.000 | 0.9798 ± 0.000 | 0.9798 ± 0.000 | 0.9646 ± 0.000 | 1.0000 ± 0.000 | 0.9818 ± 0.000 |
| DenseNet + VGG16 | 0.9848 ± 0.000 | 0.9697 ± 0.000 | 0.9747 ± 0.000 | 0.9646 ± 0.000 | 1.0000 ± 0.000 | 0.9788 ± 0.000 |
| DenseNet + VGG16 + ViT-Large | 0.9848 ± 0.000 | 0.9646 ± 0.000 | 0.9747 ± 0.000 | 0.9798 ± 0.000 | 1.0000 ± 0.000 | 0.9808 ± 0.000 |
| DenseNet + ViT-Large | 0.9848 ± 0.000 | 0.9747 ± 0.000 | 0.9747 ± 0.000 | 0.9697 ± 0.000 | 0.9697 ± 0.001 | 0.9747 ± 0.000 |
| Inception-v3 | 0.9848 ± 0.000 | 0.9747 ± 0.000 | 0.9747 ± 0.000 | 0.9596 ± 0.000 | 0.9848 ± 0.000 | 0.9758 ± 0.000 |
| Inception-v3 + ResNet50 | 0.9848 ± 0.000 | 0.9848 ± 0.000 | 0.9747 ± 0.000 | 0.9596 ± 0.000 | 0.9949 ± 0.000 | 0.9798 ± 0.000 |
| Inception-v3 + ResNet50 + VGG16 | 0.9848 ± 0.000 | 0.9798 ± 0.000 | 0.9747 ± 0.000 | 0.9697 ± 0.000 | 1.0000 ± 0.000 | 0.9818 ± 0.000 |
| Inception-v3 + ResNet50 + ViT-Large | 0.9848 ± 0.000 | 0.9899 ± 0.000 | 0.9747 ± 0.000 | 0.9697 ± 0.000 | 1.0000 ± 0.000 | 0.9838 ± 0.000 |
| Inception-v3 + VGG16 | 0.9848 ± 0.000 | 0.9697 ± 0.000 | 0.9697 ± 0.000 | 0.9697 ± 0.000 | 1.0000 ± 0.000 | 0.9788 ± 0.000 |
| Inception-v3 + VGG16 + ViT-Large | 0.9848 ± 0.000 | 0.9848 ± 0.000 | 0.9747 ± 0.000 | 0.9596 ± 0.000 | 0.9848 ± 0.000 | 0.9778 ± 0.000 |
| Inception-v3 + ViT-Large | 0.9848 ± 0.000 | 0.9697 ± 0.000 | 0.9798 ± 0.000 | 0.9545 ± 0.000 | 1.0000 ± 0.000 | 0.9778 ± 0.000 |
| ResNet50 | 0.9848 ± 0.000 | 0.9697 ± 0.000 | 0.9242 ± 0.004 | 0.9697 ± 0.000 | 0.9899 ± 0.000 | 0.9677 ± 0.001 |
| ResNet50 + VGG16 | 0.9848 ± 0.000 | 0.9899 ± 0.000 | 0.9242 ± 0.004 | 0.9545 ± 0.000 | 0.9848 ± 0.000 | 0.9677 ± 0.001 |
| ResNet50 + VGG16 + ViT-Large | 0.9848 ± 0.000 | 0.9697 ± 0.000 | 0.9293 ± 0.003 | 0.9697 ± 0.000 | 0.9848 ± 0.000 | 0.9677 ± 0.000 |
| ResNet50 + ViT-Large | 0.9848 ± 0.000 | 0.9646 ± 0.000 | 0.8434 ± 0.010 | 0.8586 ± 0.018 | 0.9646 ± 0.001 | 0.9232 ± 0.004 |
| VGG16 | 0.9848 ± 0.000 | 0.9747 ± 0.000 | 0.9697 ± 0.000 | 0.9798 ± 0.000 | 0.9899 ± 0.000 | 0.9798 ± 0.000 |
| VGG16 + ViT-Large | 0.9798 ± 0.000 | 0.9394 ± 0.001 | 0.8838 ± 0.015 | 0.9545 ± 0.001 | 0.9141 ± 0.015 | 0.9343 ± 0.001 |
| ViT-Large | 0.9848 ± 0.000 | 0.9747 ± 0.000 | 0.9848 ± 0.000 | 0.9747 ± 0.000 | 0.9697 ± 0.001 | 0.9778 ± 0.000 |

Table 8: Parameter variation and fraud detection consistency across 20 BO runs. $C$ is average consistency reported as *mean ± std*. Top 3 variable parameters show their standard deviations.

| Model | $C$ | $\bar{d}$ | Top 3 Variable Parameters ($\sigma_k$) |
|---|---|---|---|
| ResNet50 | 0.9263 ± 0.034 | 4.2635 | stroke_w (0.4104), font_size (0.3651), yc (0.3567) |
| ResNet50+VGG16 | 0.9574 ± 0.008 | 3.4855 | stroke_w (0.4702), font_size (0.3899), xc (0.3876) |
| VGG16 | 0.9513 ± 0.007 | 3.0829 | stroke_w (0.4702), zc (0.3669), yc (0.3307) |
| ViT-Large | 0.9149 ± 0.045 | 4.1524 | stroke_w (0.4443), font_size (0.3804), font_style (0.3697) |
| ViT-Large+ResNet50 | 0.9439 ± 0.033 | 4.4818 | stroke_w (0.4894), xc (0.3832), yc (0.3649) |
| ViT-Large+ResNet50+VGG16 | 0.9510 ± 0.011 | 3.6575 | stroke_w (0.4702), zc (0.3981), yc (0.3951) |
| ViT-Large+VGG16 | 0.9513 ± 0.020 | 3.6724 | stroke_w (0.4443), yc (0.3913), font_size (0.3600) |

both models fail to accurately replicate the font size and style, which are critical for maintaining the authenticity of identity documents. The output image dimensions from both GPT-4o and GPT-Image-1 also differ from those of the original input.

These inconsistencies suggest that although GPT-4o and GPT-Image-1 exhibit some capacity for layout replication, they lack the precision necessary to maintain the structural and semantic fidelity required for realistic ID template generation.

Table 9: Distances between normalized best-parameter vectors ($D$) using different guiding model(s). Smaller $D$ indicates more similar generation preferences. Model abbreviations: R50 = ResNet50, VGG = VGG16, ViT-L = ViT-Large.

| Model | R50 | R50+VGG | VGG | ViT-L | ViT-L+R50 | ViT-L+R50+VGG | ViT-L+VGG |
|---|---|---|---|---|---|---|---|
| R50 | 0.00 | 0.70 | 0.85 | 0.49 | 0.40 | 0.41 | 0.82 |
| R50+VGG | 0.70 | 0.00 | 0.43 | 0.64 | 0.51 | 0.44 | 0.37 |
| VGG | 0.85 | 0.43 | 0.00 | 0.66 | 0.69 | 0.61 | 0.21 |
| ViT-L | 0.49 | 0.64 | 0.66 | 0.00 | 0.37 | 0.41 | 0.64 |
| ViT-L+R50 | 0.40 | 0.51 | 0.69 | 0.37 | 0.00 | 0.34 | 0.61 |
| ViT-L+R50+VGG | 0.41 | 0.44 | 0.61 | 0.41 | 0.34 | 0.00 | 0.61 |
| ViT-L+VGG | 0.82 | 0.37 | 0.21 | 0.64 | 0.61 | 0.61 | 0.00 |

Table 10: Consistency Score (Mean ± Std) for Various Target Models W/O Guiding Models (Baseline)

|  | EfficientNet-b3 | ResNet50 | ViT-Large | Average |
|---|---|---|---|---|
| BO w/ SSIM-only objective | 0.9566 ± 0.028 | 0.6728 ± 0.131 | 0.6465 ± 0.193 | 0.7586 ± 0.172 |

Table 11: Consistency Score (Mean ± Std) for Various Target Models W/ Guiding Models (Our proposed approach)

| Guiding Models | EfficientNet-b3 | ResNet50 | ViT-Large | Average |
|---|---|---|---|---|
| EfficientNet-b3 | 0.9860 ± 0.004 | 0.7655 ± 0.166 | 0.8531 ± 0.169 | 0.8682 ± 0.091 |
| EfficientNet-b3 + ResNet50 | 0.9851 ± 0.004 | 0.9204 ± 0.023 | 0.9078 ± 0.048 | 0.9378 ± 0.041 |
| EfficientNet-b3 + ViT-Large | 0.9817 ± 0.002 | 0.8310 ± 0.118 | 0.9301 ± 0.011 | 0.9143 ± 0.077 |
| ResNet50 | 0.9779 ± 0.012 | 0.9356 ± 0.004 | 0.9301 ± 0.018 | 0.9479 ± 0.026 |
| ResNet50 + ViT-Large | 0.9782 ± 0.006 | 0.9135 ± 0.026 | 0.9247 ± 0.026 | 0.9388 ± 0.035 |
| ViT-Large | 0.9802 ± 0.008 | 0.8597 ± 0.038 | 0.9161 ± 0.037 | 0.9187 ± 0.060 |

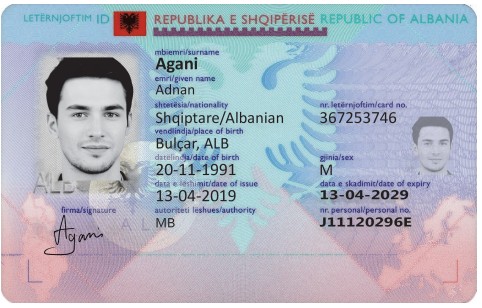

(a) Sample Image

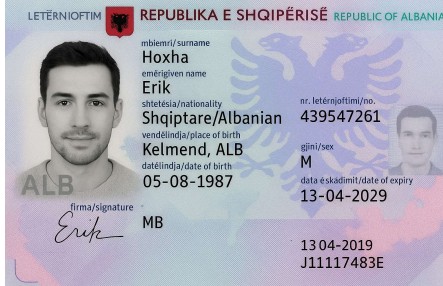

(b) Generated Image from GPT-4o

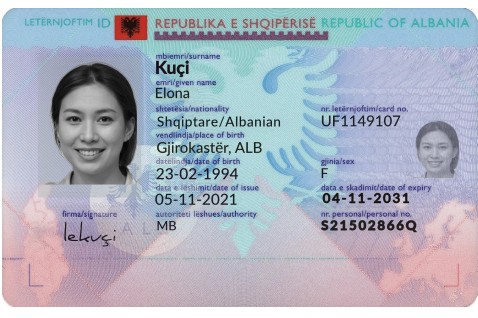

(c) Generated Image from IDSPACE

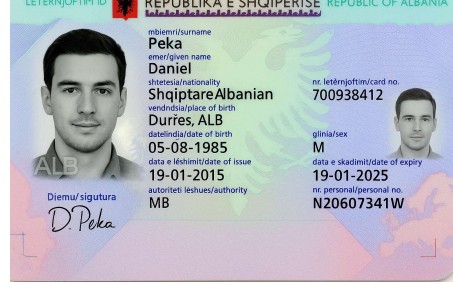

(d) Generated Image from GPT-Image-1

Figure 16: Comparison of Sample and Generated Image

## A.10 FIDELITY DISCUSSION

### A.10.1 FRAUD PATTERNS

Crop-and-move and inpaint-and-rewrite are standard fraud patterns, as discussed in existing works, such as SIDTD Boned et al. (2024). We exactly used SIDTD's methodology to generate these two fraud patterns randomly. The details are as follows:

Table 12: Breakdown of the manual effort required to introduce a new type of identity document

| Steps | Time Required (seconds) |
|---|---|
| Identify meta data | 148 ( for all 21 fields) |
| Generate template(prompt development and tuning) | 246 |
| Predefine hyperparameter for each field | 1356 |
| Configure the scripts for filling metadata to template | 1210 |

Table 13: Breakdown of the automatic processing (i.e., our scripts) required to generate documents for the new type of identity document

| Steps | Computing Time (seconds) |
|---|---|
| Stable diffusion Generate template | 16.9 |
| Generate synthetic metadata | 39.53 (for automatically generating 1000 images) |
| Hyperparameter search | 42,264 |
| Fill metadata to template | 1413 (for 1000 images) |

Table 14: Results of using GPT-4o to detect generated/diffused samples

| Methods | ACC | PRC | Recall | F1 |
|---|---|---|---|---|
| Zero-shot | $0.4667 \pm 0.058$ | $0.4167 \pm 0.144$ | $0.1167 \pm 0.02$ | $0.1801 \pm 0.04$ |
| Few-shot (2 demonstrations for real and generated respectively) | $0.5167 \pm 0.058$ | $0.5170 \pm 0.058$ | $0.5167 \pm 0.076$ | $0.5159 \pm 0.061$ |
| Few-shot (4 demonstrations for real and generated respectively) | $0.5917 \pm 0.076$ | $0.5796 \pm 0.060$ | $0.6667 \pm 0.126$ | $0.6177 \pm 0.081$ |
| Few-shot (6 demonstrations for real and generated respectively) | $0.5750 \pm 0.090$ | $0.5693 \pm 0.077$ | $0.5833 \pm 0.144$ | $0.5747 \pm 0.107$ |

(1) Inpaint-and-Rewrite Fraud Pattern. One text field is randomly selected from all the available fields (except photos and signature) on the ID. A realistic mask is then applied to this specific region containing the field, while the font size and style for the replacement text are chosen randomly from the fonts available.

(2) Crop-and-Move Fraud Pattern. In this method, a field (e.g., last name, date of birth, address, etc.) is randomly selected from one ID and then cropped and replaced with a field from another ID. In both cases, the fields are selected randomly, with a $95\%$ probability that the same field is chosen in both IDs and a $5\%$ probability that different PII fields are selected, following SIDTD.

Additionally, these two fraud patterns account for the majority of digital identity fraud. For example, the Identity Fraud Report 2024 Onfido (2023), released by Onfido (now Entrust), a top technology company on remote identity verification, reveals the following long-standing trends:

Digital fraudsters are more inclined to use an existing template, e.g., an image of a document found online, and alter it or manipulate it using digital tools.

Most of the document fraud from 2023 (80.3%) follows the above fraud patterns, which may even include document elements that are obviously wrong and visible to the trained eye, e.g., a manipulated field, or wrong printing techniques.

The fraudsters are seeking results with minimal effort and maximum reward, and achieve this by attacking in large volumes. By using scalable models to target businesses, they can determine what works before exploiting that loophole to attack en masse.

Our tool will help businesses establish a scalable fraud detection system to identify and prevent these large-scale attacks.

### A.10.2 CAN LLM DETECT IDSPACE DOCUMENTS AS GENERATED OR SYNTHETIC?

We conducted experiments to test whether existing LLM models such as GPT-4o could detect that our generated documents are generated. As shown in the table below, we tested with a zero-shot method and the few-shot method, providing $2$, $4$, and $6$ examples in the generated and real categories, respectively, but the performance of GPT-4o remained poor as shown in Tab. 14. Our synthetic data generation method is stealthy to GPT-4o.

### A.11 ABLATION STUDY: TUNING OF $\lambda_i$.

To investigate the impact of the $\lambda$ coefficients on consistency in the objective function, we conducted additional experiments on the model-guided BO method using the SIDTD dataset for the Albanian (ALB) region. We fixed the guiding model to ResNet50 and set the SSIM weight ($\lambda_0$) to 1. We then varied the consistency weight ($\lambda_1$) across a range of values: 0, 0.2, 0.5, 1, 1.5, 2, 5, and 10. The results are presented in Table 15.

Table 15: Consistency for different $\lambda_1$(Mean ± Variance)

| $\lambda_1$ | 0 | 0.2 | 0.5 | 1 | 1.5 | 2 | 5 | 10 |
|---|---|---|---|---|---|---|---|---|
| Consistency | $0.4889 \pm 0.061$ | $0.9315 \pm 0.000$ | $0.9167 \pm 0.001$ | $0.9407 \pm 0.001$ | $0.9481 \pm 0.000$ | $0.9537 \pm 0.000$ | $0.9509 \pm 0.000$ | $0.9306 \pm 0.001$ |

Table 16: Vision Models Used in Recent Academic Research

| | Models Used for Fraud Detection |
|---|---|
| SIDTD Boned et al. (2024) | EfficientNet-B3, ResNet50, ViT-large, etc. |
| Kid34k Park et al. (2023) | ResNet18, ResNet34, EfficientNet, DenseNet, etc. |
| Khare et al. (2024) | CNN, EfficientNet, etc. |

From the results, we observe that the consistency score increases as the consistency weight $\lambda_1$ increases, up to a value of 2. Beyond this point, the consistency score begins to slightly decline. This trend suggests that both SSIM and the consistency score play important roles in guiding the generation process. SSIM, which measures perceptual similarity to the target image, remains fundamental for maintaining visual quality, while the consistency term helps ensure semantic alignment. Therefore, a balanced combination of these two objectives is essential for optimal performance.

### A.12 IDENTITY DOCUMENT FRAUD DETECTION MODELS USED IN ACADEMIC AND INDUSTRY

First, this work focuses on fraud detection in documents digitally captured under white light conditions, rather than using multi-spectral imaging techniques such as near-infrared and ultraviolet light. It also focuses on a binary classification task for specific fraud patterns following a broad class of academic works in this area Boned et al. (2024). We utilized six commonly used, open-source vision architectures (ViT, ResNet, Inception, DenseNet, VGG16, and EfficientNet), each widely employed in recent academic research Boned et al. (2024); Park et al. (2023); Khare et al. (2024) and industrial fraud detection pipelines Onfido (2023); Bruveris et al. (2020); Gietema et al. (2024); Mahadevan et al. (2023); Bayer et al. (2025) for remote identity verification, as shown in Tab. 16 and Tab. 17. Our selected fraud detection models, fine-tuned on real data, serve as strong surrogates for generalizable fraud detection behavior. Our synthetic data generation method can be easily adopted in commercial black box platforms that are typically inaccessible due to IP restrictions. These platforms could use their models as guiding models to apply our approach to generate documents for model evaluation.

### A.13 MOBILE DOCUMENT GENERATION

A key application of the IDSPACE is the generation of realistic mobile document images, including photographs of identity documents captured under diverse real-world backgrounds. In this section, we present a pipeline for generating such data by replacing a document in an existing background image with a document generated by IDSPACE. The challenge in this process is to ensure that the inserted document appears natural and visually consistent within the context of the original image. This involves solving several technical problems, including accurately detecting and localizing the original document, segmenting it from the background, aligning the new document to match the original perspective, and blending it seamlessly into the scene.

To address these challenges, we employ a combination of advanced computer vision models and image processing techniques. In particular, we use Grounding DINO (Liu et al., 2024), a state-of-the-

Table 17: Vision Models in Industrial Fraud Detection Pipelines

| | Models Used for Fraud Detection |
|---|---|
| Onfido (Now Entrust) | Onfido's Atlas AI platform supports micro-model ensembles ( 10k ML models Onfido (2023)), including convolutional models Gietema et al. (2024) such as ResNet Bruveris et al. (2020), VGG16 Mahadevan et al. (2023), and ViT Mahadevan et al. (2023) backbones. |
| MicroBlink | MicroBlink's Know Your Customer (KYC) platform leverages ViT models for core platform and EfficientNet models for edge Microblink (2025). |
| Jumio | CNN-based document neural networks Bayer et al. (2025). |

art model that integrates object detection and language grounding, to detect and localize the document in the original background image. Once localized, the Segment Anything Model (SAM) (Kirillov et al., 2023) is applied to obtain an accurate segmentation mask of the original document. The prompt-based interface of SAM allows for precise and flexible segmentation, which is critical for accurate geometric alignment.

For the blending stage, we adopt the Deep Image Blending (DIB) framework (Zhang et al., 2020), which synthesizes high-quality images by optimizing a combination of loss functions including Poisson gradient loss, content loss, style loss, histogram loss, and total variation loss. To enhance structural fidelity, we extend the DIB loss with an additional differentiable Structural Similarity Index (SSIM) loss. This augmentation improves both local and global consistency between the blended image and the background.

**Parameterization.** Once we have the enhanced DIB model trained, for each generation, a user can flexibly specify to use the background of a certain existing mobile ID image (e.g., a picture of A's driver's license (DL) placed on top of the keyboard of a computer), and a template image, e.g., B's DL, to be blended by replacing A's DL in the image by B's DL. When generating a batch of documents, the users can flexibly specify the distribution of background images. For example, given a database of existing mobile documents that are annotated with labels describing the objects in the background, indoor or outdoor, lighting condition, the type of mobile phone used to capture the image, etc., e.g., MIDV's collection of mobile documents, users can specify whether the new mobile dataset to generate will focus on indoor settings or outdoor settings, or it should involve 50% of indoor images and 50% of outdoor images. Users can also specify the distribution of age, gender, ethnicity groups, and fraud patterns of the entities involved in the new template images to be blended with existing documents. For example, using our tool and a database of existing mobile documents, a user can easily generate a batch of mobile documents featuring 100 Spanish IDs for Asian Females with ages uniformly distributed from 5 to 95, with an indoor background captured by a discontinued Samsung Galaxy C5 mobile phone, for testing their newly trained models. The control parameters to be automatically tuned are the same with scanned images, as shown in Tab. 1.

Using the above methodology, we have generated 50 mobile documents for each of the ten European identity document types included in the IDSPACE dataset to illustrate the robust use case of our IDSPACE framework. Samples of the mobile documents generated for each of ten European identity document types are described in Figure 17, with background images in Figure 18. While some small issues remain to be improved, such as collecting a diverse set of background images with detailed annotations to facilitate semantic search of backgrounds to match user requirements, and addressing inconsistent sizes and lighting conditions between the document in the background image and user requirements, we believe the solutions to these problems are orthogonal with the parameterized and model-guided framework we proposed in this work, and can be addressed in our future works.

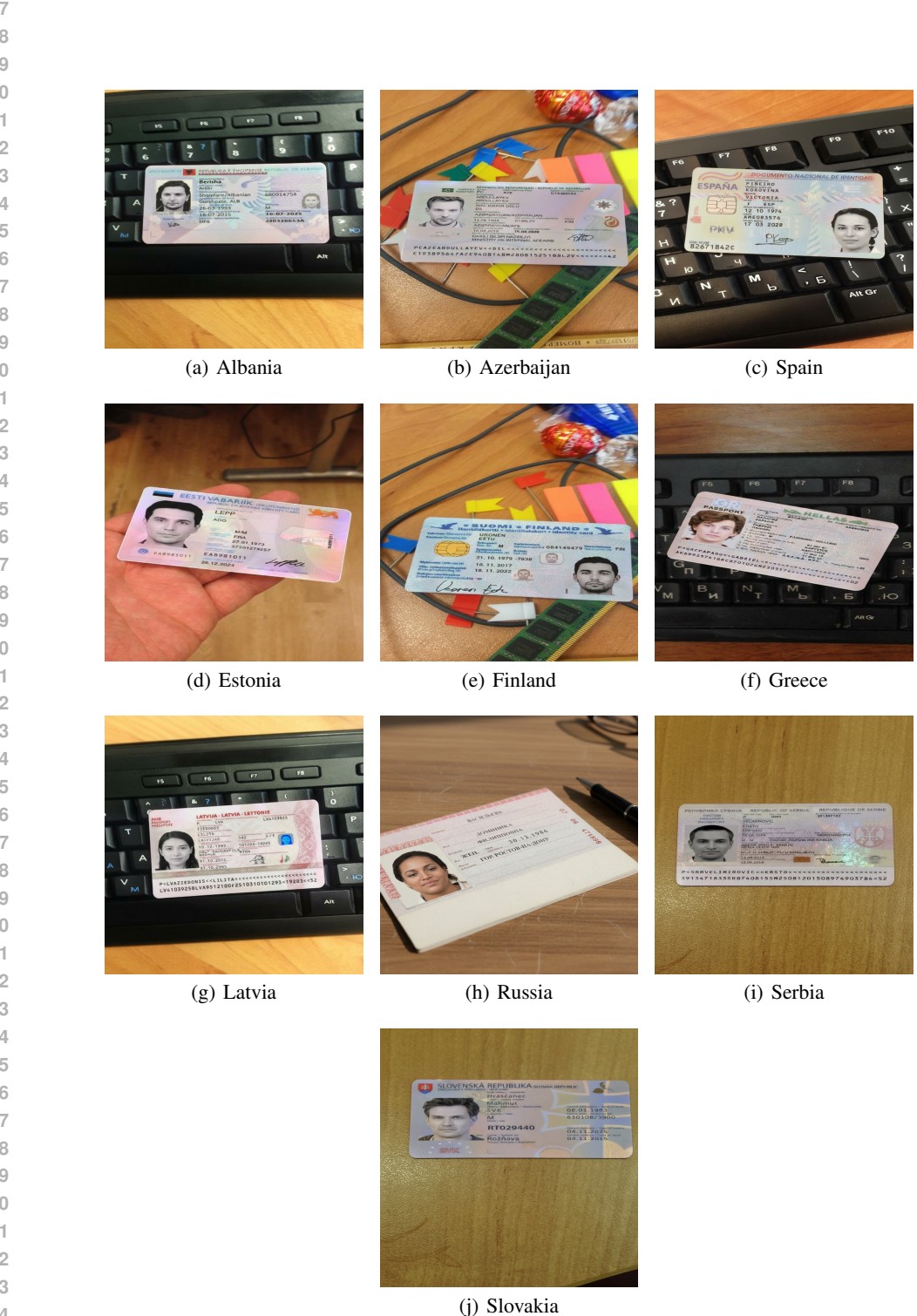

Figure 17: Example Mobile document images generated for the 10 different templates.

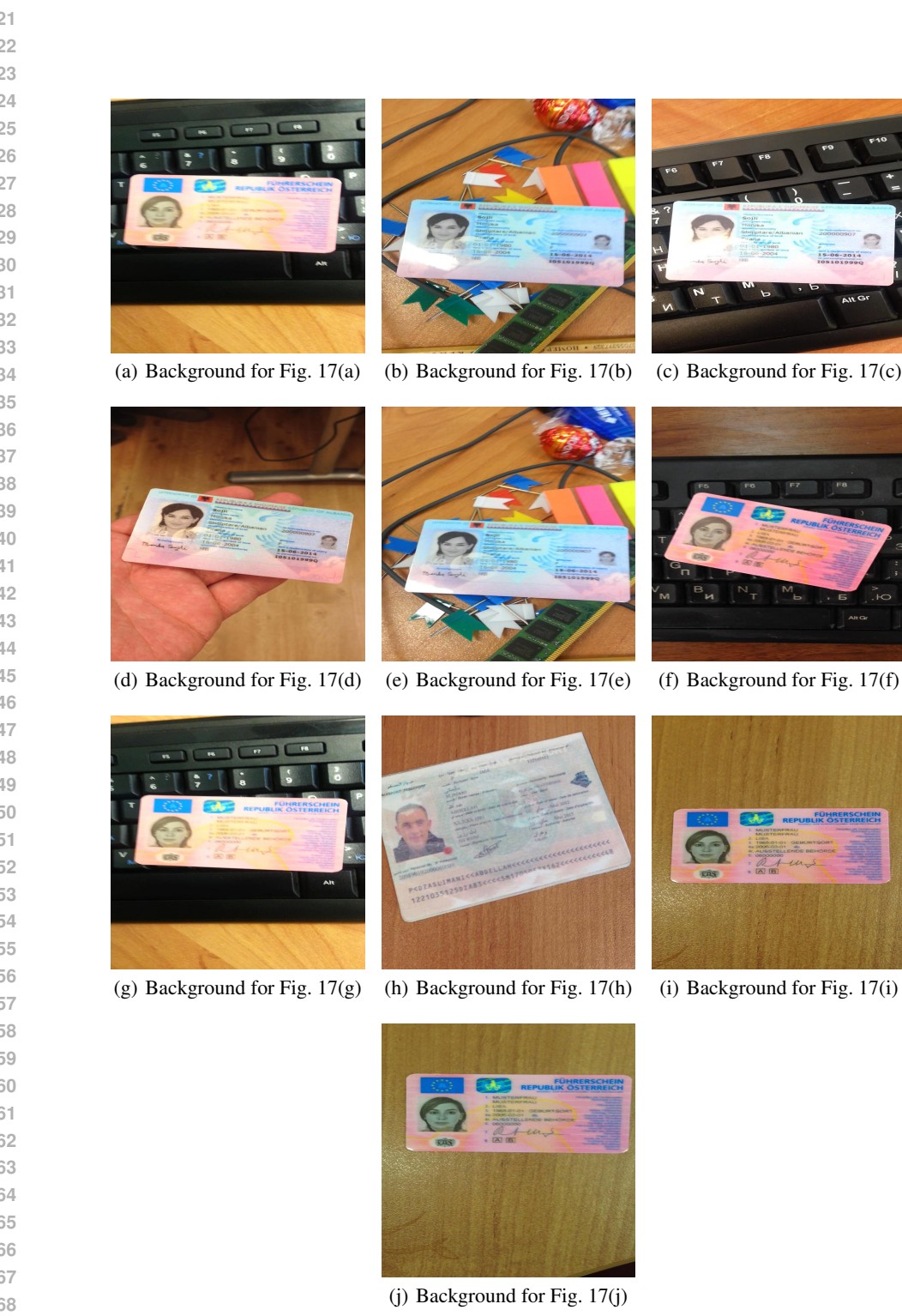

(a) Background for Fig. 17(a)  (b) Background for Fig. 17(b)  (c) Background for Fig. 17(c)

(d) Background for Fig. 17(d)  (e) Background for Fig. 17(e)  (f) Background for Fig. 17(f)

(g) Background for Fig. 17(g)  (h) Background for Fig. 17(h)  (i) Background for Fig. 17(i)

(j) Background for Fig. 17(j)

Figure 18: Background images used in Fig. 17.

