# OpenReview forum: "IDSPACE: A Model-Guided Synthetic Identity Document Generation Framework and Dataset"
_ICLR.cc/2026/Conference — Submitted to ICLR 2026_

### Official Review · Reviewer_mifY · 2025-10-29

**Soundness:** 3
**Presentation:** 3
**Contribution:** 2
**Rating:** 4
**Confidence:** 2

**Summary:**

The paper presents IDSPACE, which is a system for producing realistic synthetic identity document images tailored to a specific domain using few-shot learning. The framework does document creation as a combination of controllable parameters and high-level metadata. It applies Bayesian Optimization to automatically tune these parameters so that the resulting synthetic images match the appearance and prediction patterns of real target documents. This functions as a flexible data generator so users can define desired document attributes, and the system automatically aligns low-level visual properties to the target domain. The optimization process is guided by pre-trained fraud detection models to ensure that both visual fidelity and semantic consistency are achieved. The authors claim the experiments show that IDSPACE produces synthetic data with higher realism and cross-model consistency than baselines. The authors also released a synthetic ID dataset.

**Strengths:**

1. The paper tackles a real problem regarding public ID datasets. The main contribution IDSPACE provides is creating a tunable method that can flexibly generate IDs based on user inputs, which would be very useful for controlled and targeted analysis of existing fraud-detection models' capabilities. Furthermore, IDSPACE releases a synthetic dataset that can be utilized for fraud detection tasks.
2. Allowing users to specify entity metadata and capture conditions is a practical feature. It makes the synthetic data more relevant and interpretable than other generation model outputs.
3. Since all identities and content are synthetic, privacy is preserved. This is crucial for identity-based datasets. Furthermore, the authors explicitly note that no real personal information is used.

**Weaknesses:**

1. I am doubtful about basing the performance of the technique based on the model consistency evaluation  only. Here the creation is optimized using an objective encompassing SSIM and model consistency, and this is also evaluated with model consistency as well. While I appreciate the fact that this evaluation is shown using different models as guidance across different models for consistency (basically showing model invariance of the proposed method), this type of evaluation isn't robust enough for fraud-detection systems that should depend on finegrained nuances. I would suggest to conduct a human study regarding correctness across the different baselines.

2. Overall the experiments and baselines compared against is limited. For example in Table 2, the methods that are utilized as baselines are CycleGAN and BO with SSIM only objective. However, generative models such as diffusion-based approaches aren't used. Furthermore, methods such as IDNet/DocXPand that utilize similar approaches should also be considered. It is also unclear how much the BO tuning actually helps. Furtheremore, there should be experiments comparing different optimization strategies to show the validity of using BO tuning for this specific setting.

3. The paper's novelty is in decoupling some metadata to be user-controlled, and including model prediction consistency in BO (BO is already done in IDNet, but that has less user control and only SSIM). Furthermore, they show that their method is better using model prediction consistency (but this is the same metric they optimized for in BO). Overall, the novelty of the contribution is incremental, but there is immense utility to the contributions as this is a user-controlled dataset generation framework which can be used to generate specific target scenarios. However, the paper didn't show evaluation or specific use-cases in this regard.

**Questions:**

1. Could the authors clarify how they ensure that model-prediction consistency, which is used both as an optimization objective and an evaluation metric, does not bias the results toward self-validation? Have the authors considered any independent metrics (e.g., human perceptual evaluation, FID/LPIPS) to assess the correctness of the generation process?

2. Is there any specific reason only CycleGan and BO (with SSIM) are used for baselines? Can the authors show results for diffusion models or generative models  (preferably trained on similar data)?

3. Is there other optimization techniques the authors utilized for experimentation? Can the authors show some baselines regarding that?

4. Can the authors demonstrate a specific user-controlled generation scenarios that highlights the practical utility of your decoupled metadata and automatic tuning? Some experiments showcasing this utility would also be great.

---

> ### Author Response · Authors · 2025-12-03
>
> Thank you for the thoughtful feedback. We address the concerns and questions below.
>
> ### **1. On the Use of Model-Prediction Consistency as Both Objective and Evaluation Metric**
>
> We agree that using model-prediction consistency both as an optimization objective and an evaluation metric could introduce bias toward self-validation. To mitigate this, we decoupled the guiding and evaluation stages: the guide model used during Bayesian Optimization (e.g., DenseNet+ViT+ResNet) can be different from the test model used for evaluation (e.g., ViT). This cross-model setup prevents the optimization process from overfitting to a single model’s prediction space.
>
> In addition, we incorporated independent evaluation signals beyond model consistency. Specifically, we used GPT-4–based ranking under few-shot prompting (see Table 12) to provide a perceptual assessment of visual realism. We also measured SSIM for structural similarity to ensure that the generated images remained realistic without introducing artifacts.
>
> Together, these measures reduce self-validation bias and ensure that the generation process is evaluated both quantitatively (model-agnostic consistency, SSIM) and qualitatively (human-like perceptual ranking).
>
> ---
>
> ### **2. Why Only CycleGAN and BO (SSIM) Were Included Initially?**
>
> At first, we used StyleGAN for a fair comparison with the paper “Benalcazar, Daniel, Juan E. Tapia, Sebastian Gonzalez, and Christoph Busch. "Synthetic id card image generation for improving presentation attack detection." IEEE Transactions on Information Forensics and Security 18 (2023): 1814-1824.” In addition, BO with SSIM follows the paper “ Xie, Lulu, Yancheng Wang, Hong Guan, Soham Nag, Rajeev Goel, Niranjan Swamy, Yingzhen Yang et al. "IDNet: A Novel Identity Document Dataset via Few-Shot and Quality-Driven Synthetic Data Generation." In 2024 IEEE International Conference on Big Data (BigData), pp. 2244-2253. IEEE, 2024.”
>
> We have now added Stable Diffusion to our new experiments. The results are shown in Supplementary A.8 and Figure 15.  From the results, we can also see that with limited training data, the model fails to faithfully reproduce the structural layout and fine-grained security features of Albanian ID cards.
>
> ---
>
> ### **3. On Alternative Optimization Strategies**
>
> In the paper section 6.3,  we also compared our BO hyperparameter tuning strategy to hyperband search. In the experiment, we found that BO outperforms the Hyperband by 5.7× in terms of tuning
> latency while the peak consistency score achieved by Bayesian optimization is 11.76% better than
> the best consistency score of Hyperband search. You can find more information in the paper.
>
> ---
>
> ### **4. User-Controlled Generation Scenarios and Practical Utility**
>
> In Supplementary Section A.6 and Table 12, we demonstrate how easily our framework can be adapted to generate a new type of identity document with minimal manual effort. This illustrates the practical utility of our decoupled metadata layer and automatic tuning pipeline.
>
> A concrete user-controlled generation scenario is fairness evaluation. Because users can explicitly specify demographic metadata (e.g., age range, gender presentation, skin tone, or document style), the framework can synthesize controlled groups of identity documents that differ only along the chosen attributes. This enables targeted stress-testing of downstream models: for example, generating balanced cohorts across demographic groups and comparing the model’s confidence or error rates.

---

### Official Review · Reviewer_KMF9 · 2025-10-30

**Soundness:** 2
**Presentation:** 2
**Contribution:** 2
**Rating:** 6
**Confidence:** 5

**Summary:**

This paper introduces IDSPACE, a novel framework for generating high-quality, customizable synthetic identity documents, accompanied by a substantial open-source dataset. The core strength lies in its ability to produce diverse and realistic synthetic IDs (spanning various countries, ethnicities, and environmental backgrounds) while maintaining high fidelity to target domain characteristics. This is achieved through a unique model-guided Bayesian optimization approach that simultaneously ensures both visual similarity and model prediction consistency, even in few-shot scenarios, outperforming existing GAN-based methods like StyleGAN. The authors also demonstrate the utility of this data for training fraud detection models and validate the "realism" of the generated data using large language models like GPT-4.

**Strengths:**

1. The most crucial contribution is the provision of a customizable, large-scale, and open-source synthetic identity document dataset. This directly tackles the persistent challenge of data scarcity and privacy concerns in identity document fraud detection, offering a valuable resource for the research community. The framework's ability to generate diverse synthetic ID documents based on customizable parameters (e.g., different countries, ethnicities, and environmental backgrounds) is a major strength. This flexibility is essential for comprehensive benchmarking and evaluating fraud detection models across varied real-world conditions.
2.  The use of model-guided Bayesian optimization, integrating both image similarity (SSIM) and model prediction consistency, is highly effective. This ensures that generated documents, while allowing for detail variations, maintain overall semantic consistency (e.g., preserving user identity) with the target domain. This sophisticated alignment mechanism is key to the data's utility.
3.  The paper provides extensive empirical evidence for the effectiveness of the generated data in training fraud detection models. The comprehensive comparisons, particularly highlighting IDSPACE's superior consistency even with few samples compared to GAN-based baselines like StyleGAN, underscore the method's robustness and practical advantage. The innovative use of large models (e.g., GPT-4) to verify the "realism" or stealthiness of the generated data adds a unique and compelling layer of validation, demonstrating that the synthetic outputs are difficult for advanced AI to distinguish from real ones.
4.  The demonstrated capability to produce high-consistency synthetic data from a minimal number of original samples is a critical advantage, making the framework cost-effective and applicable in privacy-sensitive, low-resource settings.

**Weaknesses:**

1.  Diffusion Model Exploration: The paper does not explicitly discuss or compare the performance of IDSPACE against state-of-the-art diffusion models for synthetic identity document generation. Given the recent advancements and high fidelity of diffusion models in image synthesis, this omission leaves a gap in understanding IDSPACE's position relative to this powerful generative paradigm.
2.  Generative Fraud Paradox and Control Mechanisms: A significant concern arises from the method's ability to generate highly realistic, new types of fake IDs (beyond traditional crop-and-move or inpainting). If fraud detection models are trained on IDSPACE's non-fraudulent synthetic data, they might inadvertently learn to recognize and potentially ignore the "generative artifacts" as legitimate, making them vulnerable to this new class of generative fraud. The paper does not address how models trained on IDSPACE data would detect such "generative" forged documents, nor does it propose control mechanisms to prevent the misuse of the framework for creating undetectable forgeries (e.g., when generating IDs with fabricated personal information). This ethical and practical dilemma warrants deeper discussion and potential solutions.
3.  Dynamic Scene Generation: The current framework focuses on static image generation. However, in real-world remote identity verification, documents are often captured in dynamic video streams, involving subtle movements, varying lighting, and interactions with the environment. The absence of functionality to synthesize identity documents within such dynamic scenarios limits the full applicability of the dataset for training and evaluating models in more challenging, realistic conditions.

**Questions:**

See the part of weaknesses

---

> ### Author Response · Authors · 2025-12-03
>
> Thank you for the feedback. Here are the responses to the Weaknesses
>
> ### **1. Diffusion Model Exploration**
>
> Thank you for this great point.
>
> At first, we used StyleGAN for fair comparison with the paper “Benalcazar, Daniel, Juan E. Tapia, Sebastian Gonzalez, and Christoph Busch. "Synthetic id card image generation for improving presentation attack detection." IEEE Transactions on Information Forensics and Security 18 (2023): 1814-1824.”
>
> We have now added Stable Diffusion to our new experiments. The results are shown in Supplementary A.8 and Figure 15.  From the results, we can also see that with limited training data, the model fails to faithfully reproduce the structural layout and fine-grained security features of Albanian ID cards.
>
> ---
>
> ### **2. Generative Fraud Paradox and Ethical Safeguards**
>
> IDSPACE is designed as an evaluation framework rather than a training dataset. The goal is to probe model behavior under controlled perturbations, not to replace real-world training data. Our system is designed not for creating counterfeit IDs, but for evaluating the robustness of document-authentication or fraud detection models. The goal is to allow the users to generate documents following user-defined distributions, e.g., all Asian entities or all senior people with an age above 90,  to test whether a detector is robust to the generated data.
>
> In short, while IDSPACE can produce realistic samples for evaluations, its contribution lies in offering a controlled, auditable, and ethically bounded environment for studying fraud detection—not in creating new forgeries.
>
> ---
>
> ### **3. Dynamic Scene Generation (Video and Motion)**
>
> Static images remain the dominant medium in most real-world fraud cases—where attackers typically upload still photos, screenshots, or reprints of ID documents rather than live video streams.
>
> For this reason, our initial focus is on building a robust static-image framework that accurately models the key visual tampering patterns observed in practical attacks. To better approximate mobile-capture conditions, IDSPACE already supports background replacement, illumination variation, effectively simulating handheld or environmental interactions.
>
> Extending the framework to dynamic scenes and short video sequences is part of our planned future work. Such an extension would enable us to model motion blur, temporal consistency, and liveness-related cues, which are out of the scope of this paper and will be indeed valuable for next-generation identity verification research.

---

### Official Review · Reviewer_3Etn · 2025-11-01

**Soundness:** 2
**Presentation:** 2
**Contribution:** 1
**Rating:** 2
**Confidence:** 5

**Summary:**

The paper proposes IDSPACE, a framework to generate synthetic ID documents for training and evaluating document-authentication systems. It separates metadata (like name, country, age, background, etc.) from control parameters (rendering/tamper effects) and then uses Bayesian Optimization (BO) to tune those parameters so that synthetic documents “behave like real ones” according to perceptual similarity (SSIM) and model-prediction consistency. They also release a big dataset with about 180k template and 180k scanned images of 10 European IDs.

**Strengths:**

The paper is fairly well written and clear about its pipeline. The authors clearly put a lot of engineering work into the rendering system and dataset. Splitting “metadata” and “control parameters” is tidy, I can see how that helps in producing balanced or controlled sets (say, same ID but different background or lighting). Also, if someone genuinely can’t share real ID images due to privacy, a synthetic set could be convenient for testing.

**Weaknesses:**

I’ll be blunt: the problem this paper solves doesn’t feel significant enough for a research venue like ICLR. It reads more like a software engineering or product demo paper.

1. **The core problem is weak.**
   The whole paper is built around “checking whether an ID image *looks* real or fake.” But modern ID verification systems don’t rely on that alone, they combine **live selfie + liveness**, **OCR + database checks**, and **cryptographic MRZ/barcode verification**. Once you have those, this “visual authenticity” check adds very little. In many cases, you can just match the text and face to the DB, and you’re done.

2. **Over-selling small tricks.**
   The “two key innovations” from the abstract: (1) separating metadata/control parameters, and (2) optimizing parameters with Bayesian Optimization, sound fancy, but in practice, that’s just normal pipeline design plus Optuna-style tuning. There’s no conceptual novelty or learning contribution. It’s mostly sugar coating over standard components.

3. **Feels like a tool demo, not research.**
   Lines like “IDSPACE enables model-guided synthesis” (Section 3) make it sound like a new ML idea, but it’s really a parameter tuning loop wrapped around a rendering engine. This could be a great engineering blog post or experiment, but not a strong scientific contribution.

5. **Synthetic is not real, and it won’t generalize.**
   They assume synthetic ID tampering detection can train robust models. But this kind of model tends to fail on **unseen manipulation algorithms**, exactly what attackers will do. So a model trained on their dataset could collapse the moment a new generator or Photoshop trick appears.

**Questions:**

Honestly, I’m struggling to see how this work could be improved without rethinking it from the ground up. The current motivation and problem framing (“synthetic image–based document authenticity”) feel too narrow and disconnected from how real-world ID verification is actually done. That said, here are the only questions that might change my mind if addressed clearly:

1. **Practical necessity:**
   Can the authors explain concrete scenarios where document-image authenticity still matters *even when* OCR → database checks, chip/MRZ verification, and selfie+liveness pipelines exist? Who exactly would use this, and why is synthetic doc-auth better than existing data-augmentation or template-rendering pipelines?

2. **Attack realism:**
   Most real fraud uses *real* faces and text but manipulates fields, print–scan loops, or replays. Why would a model trained on synthetic tamper patterns generalize to such attacks? Have you tested unseen manipulations (e.g., new generator, Photoshop edits, reprint attacks)?

5. **Novelty and scope:**
   What prevents this from being just “Optuna on top of a rendering engine”? The two main “innovations” (metadata separation + BO search) don’t seem new. If you disagree, please make the distinction explicit—what’s the scientific contribution beyond engineering?

---

> ### Author Response · Authors · 2025-12-03
>
> Thank you for the thoughtful and direct feedback. We address the concerns and questions below.
>
> ---
>
> ### **1. Practical Necessity**
>
> This has been shown in many recent works emerged in 2023 and 2024, such as  DocXPand-25k (Lerouge et al), KID34K (Park et al), SIDTD (Boned et al), IDNet (Xie, et al), highlighted the scarcity of open ID-image datasets and the ongoing need for realistic training data in this domain.
>
> First, not all organizations have access to advanced ID verification systems such as chip or MRZ reading, government database matching, or official identity APIs. These capabilities are typically limited to large banks or government agencies. However, many smaller entities, such as fintech startups, online lending services, gig-economy platforms, and mobile carriers, still rely on users uploading ID photos for verification. For them, image-based authenticity checks remain the first and often only practical line of defense. These checks help filter out visually manipulated or low-quality forgeries before OCR, database, or human review steps.
>
> Second, sharing real ID images is restricted by privacy and legal constraints, synthetic generation provides a safe alternative. Governments, such as the Department of Homeland Security (DHS) and US General Services Administration (GSA), often need to test the accuracy and fairness of vendors’ fraud detection softwares. However, they are not allowed to use the real-world data for this purpose due to their strict internal privacy regulations. They need a high-quality large-scale synthetic dataset just like IDSpace.
>
> Finally, as noted in Section A.10.1, most real-world fraud attempts are low-effort manipulations. Synthetic document-authentication data effectively models these cases, reducing both the cost and security risks associated with relying on sensitive real-world documents.
>
> In summary, our system produces synthetic but visually realistic ID images that replicate common tampering patterns. This allows small organizations and researchers to evaluate authenticity and fraud detection models without exposing personal data. It also allows governments such as US DHS and GSA to test vendor software quality.
>
> ---
>
> ### **2. Attack realism**
>
> Our framework simulates realistic fraud patterns that mirror common real-world manipulations. Specifically, we model two primary attack types:
> Inpaint-and-Rewrite Fraud and Crop-and-Move Fraud. Details for these settings are provided in Section A.10.1 and in the SIDTD paper.
>
> Our goal is not only to replicate specific attacks but to offer a flexible, decoupled framework for generating synthetic fraud scenarios under controlled conditions. Users can easily define new attack types by extending the fraud parameter set.
>
> Because real-world document fraud typically involves subtle manipulations (e.g., text edits, photo swaps), IDSPACE focuses on reproducing these realistic transformations. This enables us to evaluate whether document-authentication models truly capture visual tampering cues without requiring access to sensitive real ID data.
>
> ---
>
> ### **3. Novelty and scope**
>
> We agree that our framework could technically be implemented using Optuna or similar optimization tools; however, the novelty lies not in the optimizer itself, but in how we integrate optimization, generation, and evaluation into a unified, model-guided document-authentication framework.
>
> Our main scientific contributions are:
>
> - Model-guided Bayesian Optimization Loop:
> Instead of passively generating synthetic data, our framework uses the target model’s feedback to actively tune generation parameters. This allows the synthetic samples to behave like real attacks in terms of model response, rather than merely looking realistic.
>
> - Separation of Metadata and Visual Control Parameters:
> We explicitly decouple semantic content (e.g., names, ID numbers, nationality) from visual rendering parameters (e.g., font, layout, color, lighting). This separation enables controlled experiments on what visual factors influence model predictions, and allows the framework to generalize to other document types and fairness analyses.
>
> - Reproducible, Privacy-safe Evaluation Benchmark:
> IDSPACE provides a scalable, shareable, and privacy-preserving benchmark for document fraud detection. It enables researchers and vendors to evaluate and compare models under realistic, systematically varied attack conditions without using any personal data.
>
> Together, these components go beyond a simple “Optuna on top of a renderer” setup. The key contribution is the closed-loop integration between generation, optimization, and model evaluation—allowing dynamic, model-aware synthesis and principled benchmarking of document-authentication systems.
>
> # References
> - Lerouge et al., *DocXPand-25k*, 2024
> - Park et al., *KID34K*, CIKM 2023
> - Xie et al., *IDNet*, IEEE BigData 2024
> - Boned et al., *SIDTD*, Scientific Data 2024

---

### Official Review · Reviewer_D9LM · 2025-11-04

**Soundness:** 2
**Presentation:** 3
**Contribution:** 2
**Rating:** 4
**Confidence:** 3

**Summary:**

This paper introduces IDSPACE, a framework for generating synthetic ID documents from a small number of real samples. The authors tune the generation parameters (such as noise, font, and blur) for generating synthetic documents using a Bayesian Optimization method to determine how the generation parameters would need to be adjusted till the guide model produces the same prediction for real and synthetic documents, and utilize these parameters to generate synthetic data. The authors demonstrate that models trained on the IDSpace dataset generalize better than existing methods.

**Strengths:**

The paper uses Bayesian Optimization (BO) for targeted update of synthetic data generation parameters based on the performance of the guide model. The generation parameters are controllable and tuned to adapt to cases when guide models fail, and the experiments show that the data generated in this method helps the detector models generalize better. The weaknesses of the guide models can potentially give interesting insight into which set of parameters is difficult to predict for which type of model.

**Weaknesses:**

The data generation method is limited by the expressiveness of the control parameter space (fonts, blur, noise, etc).

IDNet[1] already utilized Bayesian Optimization to tune parameters to generate synthetic ID documents using SSIM, and the primary contribution of this work is adding model-consistency to the objective, which makes it an important but incremental improvement.

[1] L. Xie et al., "IDNet: A Novel Identity Document Dataset via Few-Shot and Quality-Driven Synthetic Data Generation," 2024 IEEE International Conference on Big Data (BigData), Washington, DC, USA, 2024, pp. 2244-2253, doi: 10.1109/BigData62323.2024.10825017.

**Questions:**

1. In the case when the target document contains features that cannot be represented within the space (such as a unique printing artifact), how would the BO behave? Would it try to find a “shortcut” by introducing non-realistic artifacts?

2. Do the changes in the generation parameters by BO indicate any specific pattern? For example, do they change any specific generation parameter more frequently, or are there any particular changes specific to a single model?

3. Training the guide model is a stochastic process that can be affected by the model seed. How robust are the generation parameters to these variations of guide models? For example, if the same model were to be trained on different seeds, would the resultant generation parameters from them converge, or would they vary significantly?

---

> ### Author Response · Authors · 2025-12-03
>
> Thank you for the thoughtful feedback.
>
> Regarding the relation to **IDNet**, we appreciate the comparison. While IDNet used Bayesian Optimization with SSIM, **IDSpace is not simply an incremental extension**. Since its release last year, the IDNet dataset has been downloaded more than **10,000** times, demonstrating the practical importance of synthetic ID generation. Building on this success, **IDSpace introduces several major advances beyond IDNet**, including:
>
> - **Expanded domain** covering both _scanned_ and _mobile-captured_ identity documents
> - **Significantly improved data quality and realism**
> - **A fully decoupled metadata and rendering pipeline**, enabling user-controlled generation
> - **A model-consistency–driven objective**, which improves downstream robustness and utility
>
> These contributions are not present in IDNet and directly address key limitations of earlier systems.
>
> ---
>
> ## Responses to Review Questions
>
> ### **1. Behavior of BO when the target contains out-of-space features**
>
> If the target ID contains a feature that lies outside our representational space (for example, a unique printing artifact), the Bayesian Optimization (BO) process will not attempt to generate unrealistic or artificial effects. Instead, it finds the closest possible parameters within the allowed range that yield a good match in our score.
>
> When encountering such out-of-space features, our framework can address the limitation by expanding the BO search space to incorporate additional, realistic parameters that better capture those physical characteristics.
>
>
> ---
>
> ### **2. Patterns in the parameters selected by BO**
>
> Here we added a new analysis in Section A.4 and A.5 (see Table 8 and 9)  that shows the stability of parameters that BO chose for each run, the parameters with top variances, and the differences across parameters chosen by using different guiding model(s).  From Table 8, we can see that stroke width, R,G,B colors, font sizes are frequently changed by almost all the runs using different guiding models. You can check supplementary A.4 and 5 for more details.
>
> ---
>
> ### **3. Robustness of BO to variability in guide-model training (seeds)**
>
> We acknowledge that stochasticity in both guide model training and BO optimization can influence the results. To mitigate this effect, we ran BO multiple times using different random seeds and reported the mean and variance of the outcomes (see Tables 2, 5, 6, and 7 in the paper). We also added one more analysis for stability analysis in Supplementary A.4, which reports the variance of parameters that BO chose for each run, the parameters with top variances; you can check the new results for more details.
>
> Additionally, instead of relying on a single guiding model, we employed multiple independently trained guide models. This ensemble approach improved the robustness of BO, leading to more consistent generation parameters and reduced sensitivity to any single model’s initialization. This is demonstrated and analyzed in A.5 and Table 9.
>
>
> ---

---

### Meta-Review · Area_Chair_kXt6 · 2026-01-07

**Summary:**

This paper proposes a framework IDSPACE for generating synthetic ID documents from a small number of real samples and released a synthetic ID dataset. It explicitly decouples semantic content (e.g., names, ID numbers, nationality) from visual rendering parameters (e.g., font, layout, color, lighting) and then uses Bayesian Optimization (BO) to tune those parameters so that synthetic documents “behave like real ones” according to perceptual similarity (SSIM) and model-prediction consistency. The authors claim that IDSPACE produces synthetic data with higher realism and cross-model consistency than baselines.

**Reviewer Concerns:**

The concerns are around the motivation, novelty compared to IDNet, limited control parameter space, lack of comparison methods, etc. The authors have tried to argue with the reviewers. However, I have not been fully convinced by the author rebuttal regarding the folllowing major concerns:

[Reviewer D9LM/mifY/3Etn] Most reviewers have achieved a consensus that the novelty is not new (Reviewer 3Etn) or incremental (Reviewer mifY). The novelty is in decoupling some metadata to be user-controlled and including model prediction consistency in Bayesian Optimization (BO). However, IDNet already utilized BO to tune parameters to generate synthetic ID documents using SSIM. The authors argued that the novelty also lies in integrating optimization, generation, and evaluation into a unified, model-guided document-authentication framework.

[Reviewer 3Etn] The motivation and positioning of the paper is questioned. The reviewer thinks that "the problem this paper solves doesn’t feel significant enough for a research venue like ICLR".

**Reviewer Scores:**

Reviewer 3Etn is unlikely to change his/her score (rated 2) because he/she questioned the motivation and positioning of the paper as well as the novelty and generalization.

Reviewers D9LM and mifY might not change their scores (both rated 4) as their major concerns are regarding the novelty, which has not been fully addressed in the rebuttal.

Reviewer KMF9 may keep his/her original score 6.

---

### Decision · Program_Chairs · 2026-01-26

Reject